# Process Simulation of Twin-Screw Granulation: A Review

**DOI:** 10.3390/pharmaceutics16060706

**Published:** 2024-05-24

**Authors:** Tony Bediako Arthur, Nejat Rahmanian

**Affiliations:** Chemical Engineering, Faculty of Engineering, and Digital Technologies, University of Bradford, Bradford BD7 1DP, UK; tbarthur@bradford.ac.uk

**Keywords:** twin-screw granulation, simulation, discrete element method, population balance model, computational fluid dynamics

## Abstract

Twin-screw granulation has emerged as a key process in powder processing industries and in the pharmaceutical sector to produce granules with controlled properties. This comprehensive review provides an overview of the simulation techniques and approaches that have been employed in the study of twin-screw granulation processes. This review discusses the major aspects of the twin-screw granulation process which include the fundamental principles of twin-screw granulation, equipment design, process parameters, and simulation methodologies. It highlights the importance of operating conditions and formulation designs in powder flow dynamics, mixing behaviour, and particle interactions within the twin-screw granulator for enhancing product quality and process efficiency. Simulation techniques such as the population balance model (PBM), computational fluid dynamics (CFD), the discrete element method (DEM), process modelling software (PMS), and other coupled techniques are critically discussed with a focus on simulating twin-screw granulation processes. This paper examines the challenges and limitations associated with each simulation approach and provides insights into future research directions. Overall, this article serves as a valuable resource for researchers who intend to develop their understanding of twin-screw granulation and provides insights into the various techniques and approaches available for simulating the twin-screw granulation process.

## 1. Introduction

The twin-screw granulator (TSG) is composed of two enmeshed rotating screws embedded in a barrel. This suggests a slight distinction among granulators and the prominent variations are classically subjected to design limitations, namely length and diameter of the screw and the specific screw element geometry [1]. The most common twin-screw extruders in the industry so far are co-rotating twin-screw extruders, and this is also the type that has been the most researched [2,3]. TSG is integrally a “regime-separated” granulator that enables improved regulation of the granule properties. The TSG has been modified and altered based on extrusion technology. Granules of high densification have been produced in early research works where a die plate was placed at the front end of the granulating equipment. This densification is due to the extruded material being passed through a narrow hole in the die plate. Subsequently, research has been conducted on an open configuration where the die plate has been removed to determine granule properties under varying process parameters [4]. Aspects of the TSG that have a significant influence on its mechanism and characteristics, such as the powder feed rate, screw speed, solid-to-liquid ratio, composition of the granulating liquid, method of liquid feed addition, granule properties, and process measurements such as torque and residence time, were investigated by different researchers [5]. In addition to this, studies have been conducted to better comprehend the equipment and screw geometries, such as the following [2]:Screw configuration.Conveying elements.Pitch and length, kneading elements.Angle and thickness of the screw.The screw cross-sectional area.Length to diameter (L/D).Liquid-to-solid (L/S) ratio.Properties of the material.The formulation of the binder.Screw speed.The feed rates of the material.The residence time distribution (RTD).The size distribution of the granule particles.The torque.The granule porosity/density, and the final tablet properties.

The objective of this work is to offer a thorough examination of the simulation tools and approaches used in the investigation of TSG processes. The primary contributions of this review are as follows:This text provides a comprehensive introduction to the basic ideas, design of equipment, process parameters, and simulation approaches in TSG.Examining the influence of operational parameters and composition configurations on the flow characteristics, blending performance, and particle interplay in the twin-screw granulator.This review critically examines simulation approaches, including the population balance model (PBM), computational fluid dynamics (CFD), the discrete element method (DEM), process modelling software (PMS), and linked techniques. The focus is on their application in simulating TSG processes.Analysing the difficulties and constraints linked to each simulation method and offering perspectives on potential areas for future research.

The review was conducted by systematically searching for relevant literature in databases such as ScienceDirect, Web of Science, and Google Scholar. The search phrases encompassed “twin-screw granulation”, “simulation”, “modelling”, “population balance model”, “discrete element method”, “computational fluid dynamics”, and other combinations thereof. A comprehensive search initially produced a substantial pool of prospective references, which were further evaluated based on their relevance, quality, and publication date. Emphasis was placed on peer-reviewed scientific publications, conference proceedings, and book chapters that specifically focused on simulating TSG processes. The chosen references underwent a thorough evaluation, and their discoveries were combined to create a complete summary of the present understanding of TSG simulation.

This review encompasses a broad spectrum of simulation techniques and methodologies utilised in TSG procedures. The review covers the essential principles, design of equipment, process parameters, and simulation approaches, offering a comprehensive perspective on TSG simulation. Nevertheless, due to the dynamic and ever-changing nature of this sector, it is possible that this evaluation may not encompass all the latest advancements or specialised uses. However, a deliberate and focused attempt have been made to incorporate the most pertinent and influential research in TSG modelling, guaranteeing that the review serves as a valuable and useful tool for academics and practitioners in this area of study.

Computational modelling and simulation techniques are now essential instruments for comprehending, enhancing, and expanding TSG [6,7]. Simulations offer valuable understanding of the mechanisms involved in granulation, allowing for the accurate prediction of granule characteristics, and assisting in the design and control of the process [8,9]. This study analyses the primary simulation methods used in TSG, such as population balance modelling, computational fluid dynamics, discrete element technique, and process modelling software. It focuses on their range, capacities, and constraints.

## 2. Twin-Screw Granulation

The most complicated invariable aspect of the process so far is the design of the screw element. This is because it can be pinned on the uniqueness of the individual screw elements designs, as well as the numerous variations associated with screw elements, such as the conveying, kneading, and comb mixer elements, therefore making the screw configuration have indefinite possibilities. These studies and the resulting observations accentuate the importance of the resulting granule properties as a function of screw configuration [10].

Gamlen and Eardley, in 1986, were the first pioneers to work with a twin-screw extruder for the purposes of granulated paracetamol extrusion. Lindberg et al. produced several research papers to determine what effect the process parameters and residence time have on the attributes of the granules produced and the equilibrium conditions. They achieved this by employing an extruder similar to the one used by Gamlen and Eardley [11] between the years 1987 and 1988. A patent was issued by Ghebre-Selassie et al. and Burgschweiger and Tsotsas [12] for using a TSG in a continuous granulation process in a single pass.

At present, there are several TSG in existence, varying in size and geometry. Figure 1 presents a schematic diagram illustrating the typical layout and key components of a twin-screw granulation system, offering a comprehensive overview of its design and functionality. The experimental stage granulators have made way for a more modified version to be implemented in the industry with bespoke and custom designing, i.e., the GEA Pharma Systems ConsiGma, for the purpose of continuous tableting. Since there are varying process capacities and measurements, the screw L/D is mostly used to identify granulators [2]. However, the scaling up of granulations is not linearly based upon the analogous L/D ratios but rather, its optimisation is based on the granulator. One significant benefit of the TSG has been its demonstrated effectuality in granulating high drug load formulations [13].

### 2.1. The Liquid-to-Solid Ratio (L/S) of Twin-Screw Granulation

Even though significant strides have been made in the comprehension of TSG, much work still needs to be performed in the production of high-quality granules, as this process is still not well understood. The downstream process experiences some level complexity due to the characteristic bimodal granule size distribution. Similarly, monomodal distribution granules formed at high L/S are consistently too large for tableting without milling. Fines are often abundant and factors leading to a reduction in fines often result in higher proportions of oversize agglomerates. Further work has been suggested for process optimisation, in order to increase the yield and to develop the understanding of how to achieve this [2].

The liquid-to-solid ratio is a crucial factor in twin-screw granulation. The minimum ratio of liquid to solid that is needed to steadily granulate a formulation in the twin-screw granulators is quite small compared to the liquid/solid ratio needed for conventional wet granulation techniques, like high shear mixing and fluidised bed granulation. However, the TSG can accept higher liquid-to-solid ratios than high shear granulation and can have its over-wetting point reach a higher level than that of high shear granulation. This allows the TSG to operate at a broader level, providing higher range process control. There is minimum threshold of liquid that is needed for granulation to occur; likewise, there is a maximum limit above which granulation stops and the powder turns to paste due to over wetting [14]. 

Several studies have investigated the effect of the liquid-to-solid ratio on the shape of the granules produced by twin-screw granulation. It has been established that the aspect ratio of granules decreases with increasing liquid-to-solid ratio as particles become more rounded. Granules produced in twin-screw granulators tend to be elongated with rough surfaces when the liquid-to-solid ratio is low [15]. In TSG, an increase in the viscosity of the binder liquid, which is the binder concentration, decreases the amount of liquid required to produce granules with a monomodal size distribution [16].

There is a link between the binder viscosity and granule size which depends on formulation and may need a means of optimising the material components. The viscosity of the liquid has less influence as far as the shape of the granule is concerned; however, at higher liquid viscosity, the shape of the granules becomes more rounded [17]. Several researchers found that the concentration of the liquid with regards to the shape of the granules is solely dependent on the conveying screw elements [18].

Throughout the compaction and consolidation phases of the twin-screw granulation process, the granules experience significant levels of pressure and shear stresses. The interplay of these forces prompts the particles within the granules to reconfigure and converge, diminishing the gaps between them and augmenting the overall density of the granules [19].

As the particles are brought closer together, the liquid bridges connecting them become more compressed and concentrated. Compression of the liquid bridges leads to an increased capillary force and enhanced inter-particle adhesion [20]. The enhanced potency of the liquid linkages enhances the general cohesion and stability of the granules.

Furthermore, the decrease in gaps between particles and the compaction of the granules result in a reduction in their capacity to absorb and hold fluids. The denser composition of the granules restricts the amount of space for liquid to infiltrate and be soaked up [21]. Consequently, the absorption of the granules diminishes as they become more compact and solidified.

The granules’ mechanical strength and integrity are improved by the combination of stronger liquid bonding and less inter-particle space [22]. The increased density and lower absorbance of the granules suggest the development of stronger and more stable granular structures.

Increasing the viscosity of the binder liquid can actually enhance the strength of granules; however, the underlying mechanism is more intricate than merely generating additional bonds. The primary variables involved are the rheological characteristics of the binder and their influence on the powder particles [3,23].

Recent research has indicated that augmenting the thickness of the binder can result in the creation of denser and more enduring liquid connections between the particles. This, in turn, enhances the strength of the granules [24]. Nevertheless, the heightened thickness of the granulating liquid can also cause a reduction in its ability to distribute evenly, resulting in uneven wetness and the creation of specific areas that are excessively wet [19].

Nevertheless, the heightened viscosity of the binder might potentially yield adverse consequences for the granulation process. A significant problem arises from the reduced ability of the high-viscosity granulating liquid to disseminate [25]. The diminished capacity of the binder to evenly disperse throughout the powder bed might lead to uneven wetting and the creation of specific areas that are excessively moist [13,26].

The reduced capacity of the high-viscosity binder to spread can result in the creation of big, compact granules that vary in size [27]. The increased strength of these granules is attributed to the presence of thicker liquid bridges. However, the granules’ uneven size and density can create difficulties during further processing steps, such as tableting or capsule filling [28].

An alternative method involves modifying the process parameters, such as the ratio of liquid to solid, the rate at which the binder is added, and the arrangement of the screw, in order to counteract the reduced ability of the thick binder to spread [29]. By precisely regulating these parameters, it is feasible to attain a more consistent wetting and dispersion of the binder, resulting in the creation of granules that possess the intended characteristics.

Although increasing the viscosity of the binder can enhance the strength of the granules, it is not primarily attributed to the generation of additional bonds. The rheological characteristics of the binder and its interaction with the powder particles are of utmost importance. Addressing the reduced spreadability of binders with high viscosity is crucial. This can be achieved by optimising binder formulation and adjusting the process parameters. These measures ensure the production of granules with the necessary strength and homogeneity.

### 2.2. Screw Speed

It has been found in studies that screw speed has minimal relevance in regards to the attributes that twin-screw extrusion produces. Seemingly, this is a deviation from the argument that the speed of the screw is vital in defining the fill level, which is important in establishing granule characteristics [30].

The configuration of the screw is dependent on the type of material being granulated. As the TSG is an assembly of individual elements of the screw, this makes reconfiguration possible for a specific material at any given granulation time [31]. For the twin-screw granulator, the speed at which the screw rotates has been identified as having little effect on the properties of formulated granules. However, this assertion is contrary to the claims that, in determining the barrel fill level, the screw speed is a determining factor which, in turn, is a critical parameter for granule property determination. This contradiction may be due to the fact that during normal operation levels, screw speed has a limited effect on the properties of the granules but turns out to be more influential at the lower and upper limit of the barrel fill. This is because the granule properties become more dependent on the barrel fill level. At greater conveying capacity in the granulator, because of a higher screw speed, the residence time becomes shorter. At lower speed rates, the torque values begin to increase when the feed rate is constant; thus, high torque values are a result of the granulator being filled with heavier loads of the granulating material. However, at a higher speed rate of the screw, the barrel fill is lowered as a result of the conveyor capacity being increased. This allows materials to leave the granulator at a faster rate, thus reducing the mass load and the torque. The frictional resistance is less relevant for a screw rotating at a high speed [2].

### 2.3. Mixing and Residence Time

Residence time is the time needed for the material to be transported through the equipment along the streamline. However, because of the velocity profiles, which are of non-Newtonian nature, and the different elements of the screw in the equipment, there exists a varying length of residence time, distributed at various sections of the screw elements. The RTD is dependent on the speed rate, configuration of the screw, and processing conditions. At an elevated temperature, a higher RTD becomes a major concern as this means the granulating materials become exposed to such temperatures. This could mean that the liquid binders composed of polymers, which are of high volatility, could be in danger of being destroyed along with their additives because of thermal degradation [31].

For a specific powder formulation process, there would be the need to ensure that nucleation, growth, wetting, and consolidation taking place within the extruder are happening at the appropriate residence time limits for effective granulation [10]. The residence time distribution is a vital function for polymer granulating processes taking place in a twin-screw extruder, due to the temperature issues that were discussed earlier [31]. The rotation speed of the impeller is a contributing factor to the residence time of a granulator [10]. Typically, it could be said that for a proper mixing distribution of an extrusion, average residence times must be higher; however, granulation design complexity becomes higher due to the complicated rheological mixtures used, thus making this suggestion difficult to achieve. Better knowledge of RTD is an advantageous tool in the design process of high-quality production while also achieving design criteria [32]. The recording of RTD can be conducted either by stimulus or direct test measurement. The particle tracking by use of a dye is a good example of direct test measurement [2].

Ideally, the capacity and the residence time in the barrel should be large and short, respectively, while keeping the granule integrity after going through the mill screen [33]. However, the residence time for hard materials needs to be long enough to be able to generate both coarse and fine particles [34].

Analyses of mixing must have a velocity field of high accuracy, as all analysis depends on this to produce a precise result. The complexity of obtaining an accurate velocity field arises in two major folds when balance equations are used to find the right velocity field in these complicated geometries. The first relates to the moving geometries, the screws in a fixed barrel, and the very small size of the gap widths. The rotational time change is the second challenge. Analysis of particle tracking depends heavily on the field velocity and its associated tightly fitted moving screws; obtaining these parameters is not inconsequential but rather imperative. Despite all the recent advancements made in using simulation techniques to accurately determine the twin-screw field velocity, the problem still exists when such a determination is made in real-life granulation [35].

### 2.4. Conveying Elements

The conveying parts play a vital role in the twin-screw granulator by transporting and mixing materials along the barrel [2]. The screws are constructed with helical flights that coil around the shaft, forming a channel-like gap between the flights [29] as shown in Figure 2. The spinning of the screws causes the material to move forward due to the action of the flights, which applies a mix of rotational and axial forces [28]. The conveying elements are distinguished by their pitch, which refers to the space between two successive flights [2]. The pitch may fluctuate based on the particular design and the desired features of material flow. Elements with a greater pitch enable increased material flow and faster conveyance, whereas those with a smaller pitch facilitate more thorough mixing and longer residence durations [29].

The conveying elements are the most over-relied upon element of the screw. Mixing is also an important feature of the conveying elements, as well as the transportation function, where transportation is performed by drag flow and generated pressure that carries the materials past the kneading and the die. The length of the pitch and the widths of the gaps within the screw and the barrel determine the rate of transportation and mixing; however, screw speed is also a factor [35]. Material transportation by the conveying elements is either in the forward or reverse direction. They come in different pitch lengths, with the flight length along the element being the base. The slowest conveying ability belongs to the conveying elements with the shortest pitch lengths and those where the degree of channel volume is filled the highest. Narrow-pitch conveying elements exhibit the slowest conveying capacity and the highest degree of fill of channel volume. Narrow pitches are mostly deployed in the feed zone of the extruder [36].

The conveying elements are responsible for drawing the material products into the extruder, transporting them to the initial processing site, and compressing them as and when it is required. It is also responsible for the transportation of products from one processing zone to the other, creating the retention time needed for devolatilisation, and lastly, generating the pressure that is required at the end of the extrusion to push the final products through the discharge tunnel. “The resultant effect of the conveying element is dependent on the frictional characteristics between the screw surface and the product.” This is the screw feeders law that all drag flows outside the apex obey [37].

Extrusion screws are designed with pitches of varying length to offer different conveying capabilities. Figure 2 shows the conveying element geometry where granulating products traverse along the pitch helically around the screw [36]. The conveying elements with high pitch allow for the incorporation of high volumes of material at the feeding section, thus increasing the rate of production [31]. The transportation of materials is effective when the conveying element has a large pitch element. However, a small pitch element would be more desirable when the traversing material must be pressurised [38].

Materials are affected by the conveying element in the three following key areas: conveying materials in the barrels, compaction axial generation, and the breaking of large lumps [37]. Kitching et al. (2020) [39] suggest that the large granules that form along the conveying elements would have weak strength. This also highlights the fact that the conveying elements allow for the development of the seeding process with some level of low shear. 

### 2.5. Kneading Blocks

Kneading elements are responsible for applying the needed mechanical energy (high shear) to the material using dispersive and distributive mixing [40]. Distributive mixing is the process whereby the physical distance separating two granulating materials is reduced through blinding with the aim of facilitating the occurrence of a chemical reaction or diffusion [35]. The morphologies of the materials do not change after undergoing division and recombination. The materials are evenly distributed, and the resulting particles do not break down. On the other hand, “dispersive mixing breaks up the mixture’s mirror particle into droplet” [41]. Dispersive mixing breaks the force of cohesion that exists within the agglomerates, thus leading to morphological change and size reduction in the material [42]. The screw configurations built for a classical screw element include the following: forward and reverse conveying elements, mixing elements, and kneading elements for distributive and dispersive mixing purposes [31].

Kneading discs are placed at a specific angle (examples; 30°, 45°, 60°, 90°) to form the kneading elements. The width and offset angles of the discs dictate the mixing performance of the kneading elements [43]. The transportation force in the forward flow and backflow reduces and increases, respectively, with an increasing offset angle due to the widening of the opening between the discs. Thus, the conveying capacity decreases with increasing offset angles while the compressive intensity and created shear stress are much higher, which improves the performance of the mixing [44].

Another geometry of the kneading element that plays a vital role in the mixing performance is the disc width. By using kneading discs of smaller width, the shear in the gaps of the kneading elements reduces since the particles flow around the discs more freely. While kneading discs of increasing width decrease distributive mixing because of the material flow dividing into fewer streams, the dispersive mixing intensity increases. This is due to the susceptibility of the material to circulate in the axial direction over the crest region as a result of both the downstream and the upstream forces of the discs [45].

### 2.6. Comb Mixer Elements

The twin-screw granulators that have screw mixers are a unique type of extruder. They are mostly non-intermeshing and counter-rotating. JSW, Farrel, and Teledyne (UK) are the major manufacturing companies of such machines. The conveying sections of the screw, which are helical in design, are for the transportation of the feed material to the kneading section of the system [31]. The combing elements are conveying elements, and their flights feature a succession of cuts that contribute to their distributive mixing. The combining elements’ mixing and conveying function falls between the kneading and the conveying element. The combing elements require less mechanical shear stress to mix the materials. Their ability to expedite the distribution of the liquid binder onto the solid powder bed make them the favourable option in twin-screw wet granulation. However, for the purpose of melting of some peculiar binders applied in TSMG, they might not be able to generate the needed shear for mixing [36].

### 2.7. Fill Level

Fill level is a fundamental consideration when undertaking a scale-up, primarily because its behaviour may be different despite similar screw speeds and feed rates [2]. It is necessary to have cognisance of the fill level relative to operating conditions when analysing and interpreting the final properties of granules. Little to no work has been conducted on the quantifiable determination of fill level in the literature. This is due to the complexity of calculating free volume and determination of residence time. The fill level is an important factor which should be considered in the comparison of different granulators [46].

## 3. Modelling and Simulation of Wet Granulation 

Twin-screw granulation is now a crucial procedure in the powder processing industries and the pharmaceutical sector for manufacturing granules with controlled qualities [2,29]. Twin-screw granulators offer superior control over granular characteristics in comparison to alternative granulation techniques [47]. They are extensively utilised to produce granulated intermediate and end products in several industries, including pharmaceuticals, food, agriculture, and specialised chemicals [48]. The twin-screw granulation process utilises the interplay of raw materials, equipment design, and process parameters to convert powders into granules [49]. Computational modelling and simulation techniques are now essential for comprehending, optimising, and expanding twin-screw granulation [50]. Simulations offer valuable insights into the mechanisms of granulation, allowing for the prediction of granule properties and assisting in the design and control of processes [9]. This paper analyses the primary simulation methods utilised in twin-screw granulation, such as population balance modelling, computational fluid dynamics, discrete element technique, and process modelling software. It focuses on the extent of their applicability, their capabilities, and any limits they may have.

### 3.1. Approaches to Modelling of Wet Granulation Processes

There are two main patterns for the mathematical modelling of wet granulation processes. These are physically based models and data-driven models [51].

#### 3.1.1. The Data-Driven Models (Empirical)

These models use time series data of real systems based on the input and output date at a specified time range. The model is developed by choosing an appropriate model structure and applying the model parameters to the data to achieve the best fit for the model. There are diverse ways to achieving the best fit which include altering the parameters of the structure and the parameters of the models. This type of modelling is significantly limited and is only applicable to systems within the intervals set. It is also useful when a model is required for the purpose of controlling the application; however, looking for insights into the model is not necessary.

#### 3.1.2. The Physical (Mechanistic)-Based Models

These models integrate the model’s primary concept of the chemistry and the physics behind the process. The conservation and constitutive facets are the two main aspects covered by physical-based modelling and it includes the following:Application of thermodynamic conservation principles for mass, energy, and momentum.Development of appropriate constitutive relations that define the intensive properties, mass, and heat transfer mechanisms as well as the particle growth and breakage mechanisms.Application of population balances that track particle size distributions as various particulate phenomena take place.

Mechanistic models are a very laborious and complicated process of developing a model as compared to the development of empirical models. This model is useful when a full understanding of the relevant constitutive relations is accessible, and it is useful for wider variety of applications.

Twin-screw granulation is a very promising technique in the pharmaceutical sector for continuous processing. It has several advantages compared to typical batch granulation processes [2,13]. However, in order to completely achieve its maximum capability, it is important to have a more profound comprehension of the fundamental mechanisms and the influence of process parameters on the features of granules [2]. This review article aims to examine different modelling methodologies and findings about TSG. It will serve as a basis for future study and efforts to optimise the process. The pharmaceutical sector may overcome its issues, boost product quality, and improve its worldwide competitiveness by furthering the understanding and use of TSG [52].

### 3.2. Modelling through Population Balance Model (PBM)

The PBM is a type of physical-based modelling that monitors the transformation of particles through the granulation mechanism. The PBM, in either its one-dimensional or multi-dimensional form, has been utilised to forecast the progressive alterations in granule characteristics, including granule size distribution, moisture content, density, and porosity [6]. Equation (1) represents the general equation for the PBM, Equation (2) illustrates the concept of particles in a specific region of the particle phase, and Equation (3) provides a generic depiction of the super-structure of the PBM [53].
(1)$$\frac{\delta n}{\delta t}+\nabla *\left(vn\right)-B+D=0$$
where v is the particle velocity, n is the number of density particles, with the particles’ birth rate and death rate being B and D, respectively.
(2)$$\begin{array}{cc} \{density\;function & change\;inclass,\;location\;and\;time\}\\ = & \left\{disperse\;in\;through\;boundary\right\}\\ - & \left\{disperse\;out\;through\;boundary\right\}+\left\{flow\;in\;boundary\right\}\\ - & \left\{flow\;out\;of\;boundary\right\}+\left\{grow\;in\;from\;lower\;classes\right\}\\ - & \left\{grow\;out\;from\;current\;class\right\}\\ + & \left\{birth\;due\;to\;coalescene\right\}-\left\{death\;due\;to\;coalescence\right\}\\ + & \left\{breakup\;in\;from\;upper\;clases\right\} \end{array}$$

### 3.3. DEM Simulation

The granulation process has been extensively studied in recent years through the use of discrete element method (DEM) modelling and simulation. This progress has been made possible by the availability of free open platforms like OpenFOAM and LIGGGHTS, as well as commercial software packages such as Rocky, Altair EDEM, Berker 3D Three particle/CAE, and CADFEM. The discrete element method (DEM) has been employed to examine the optimisation of processes, the development of models, and the formulation of materials in several types of granulators, such as fluidised bed, high shear, drum, and twin-screw granulators.

The discrete element method (DEM), which was pioneered by Cundall and Strack in 1979 [54] and built upon the earlier research of Alder and Wainwright in 1956, has found extensive application in the modelling of rock engineering problems, as shown by Boon et al., (2011) [55], and the resolution of issues related to granular and discontinuous materials [56]. When modelling granular matter, there are two primary methods: the continuum (Eulerian) approach and the discrete (Lagrangian) approach. The discrete approach is favoured due to its consideration of granular material as a composite of individual idealised single particles, which is more accurate compared to the continuum form [57]. 

The discrete approach can be executed using two primary techniques: the soft-sphere and hard-sphere methods. The soft-sphere technique assumes that particles are rigid, and that any deformation is represented by a small overlap between particles when they come into contact. The hard-sphere technique implies that particles are stiff and that the only way momentum is exchanged is through collisions between particles. Additionally, it considers the forces of interaction between particles to be impulsive, as described in [54].

#### Base Models

The Equations (3) and (4) describe both the translational and rotational motions of each particle, as previously stated. The equations are numerically integrated using explicit time integration strategies, such as the forward Euler or velocity Verlet algorithms [3,58].
(3)$$\frac{m_{i}\left(dv_{i}\right)}{dt}=F_{C,i}+m_{i}g\;*V$$
(4)$$I_{i}\frac{d\omega_{i}}{dt}=T_{i}$$

Let *m_i_* represent the mass of particle i, and *I_i_* represent the moment of inertia of particle i. $V_{i}\;$represents the velocity of translation, *g* represents the acceleration due to gravity, $\omega_{i}$ represents the angular velocity, $F_{C,i}$ represents the combined force of contact between particle *i* and nearby particles or walls, and $T_{i}\;$ represents the total torque applied to particle *i*.

The contact forces between particles are determined by the utilisation of several contact models. The Hertz–Mindlin model, the linear spring-dashpot (LSD) model, and the Johnson–Kendall–Roberts (JKR) model [59] are the most-often utilised models. The Hertz–Mindlin model combines Hertz theory to calculate the normal force ($F_{n}$) and Mindlin theory to calculate the tangential force $(F_{t}$).
(5)$$F_{n}=-k_{n}\;x_{n}-\eta_{n}\frac{dx_{n}}{dt}$$
(6)$$F_{t}=-k_{t}x_{t}-\eta_{t}\frac{dx_{t}}{dt}$$

$k_{n}$ and $k_{t}$ represent the spring stiffness in the normal and tangential directions, respectively. $\eta_{n}$ and $\eta_{t}$ represent the damping coefficients in the normal and tangential directions, respectively. $x_{n}$ and $x_{t}$ represent the overlaps in the normal and tangential directions, respectively.

The Hertz–Mindlin model combines the Hertz theory, which deals with normal contact, and the Mindlin theory, which deals with tangential contact. This model considers the elastic deformation of particles and is appropriate for simulating rigid, non-sticky particles. Equations (3) and (4) provide the values for the normal and tangential forces, respectively. The spring stiffnesses ($k_{n}$ and $k_{t})$ and damping coefficients ($\eta_{n}$ and $\eta_{t}$) depend on the particle properties, including Young’s modulus, Poisson’s ratio, and the coefficient of restitution [60].

The DEM is a flexible tool used to simulate particulate systems, offering important insights into the intricate behaviour of granular materials. Due to growing computing power and the advancement of algorithms, DEM is anticipated to have a vital impact on the model design and optimisation of particle processes across many industries.

### 3.4. CFD Simulation

Computational fluid dynamics (CFD) is a method that uses numerical solutions to analyse the movement of fluids and the transportation of substances. CFD, or computational fluid dynamics, is widely utilised in engineering research and industries to address and evaluate issues pertaining to heat transfer, biological engineering, aerodynamics, flight testing, weather simulation, and system design and analysis across diverse sectors [61].

The origins of modern fluid dynamics can be traced back to Leonardo Da Vinci (1452–1510). Subsequently, diligent scientific scholars have devoted significant effort to developing a mathematical description for the phenomenon. Claude Louis Marie Henry Navier and George Gabriel Stokes proposed the Navier–Stokes equation in the early 1900s. They modified the Euler equations by including viscous transport [61].

#### 3.4.1. Equations of Conservation

The conservation equations are the primary governing equations of fluid dynamics. These equations encompass the continuity equation (which accounts for mass conservation), the momentum equation, and the energy conservation equation [62].

##### Mass Conservation (Continuity Equation)

The continuity equation is the most familiar term for mass conservation. The mass conservation equation, commonly referred to as the continuity equation, is derived from the fundamental premise that mass is conserved and cannot be generated or lost during a chemical reaction. Put simply, the mass of a system remains unchanged as time progresses. The mass conservation equation is expressed in a general form as follows [63]:(7)$$\frac{\partial \rho }{\partial t}+\nabla *\left(\rho V\right)=0$$
where *ρ* and *V* are the density and volume of material. 

##### Momentum Conservation

The principle of conservation of momentum states that the total momentum of a given system remains constant. Momentum cannot be created or destroyed, rather it can only be modified by external forces in accordance with Newton’s laws of motion [63]. The principle of momentum conservation asserts that an object or a collection of objects in motion will preserve its overall momentum, defined as the product of mass and vector velocity, unless influenced by an external force. In an isolated system, such as the cosmos, where external effects are absent, momentum remains conserved. Momentum conservation ensures that its components in all directions remain conserved. The law of conservation of momentum is crucial in solving collision situations [64]. The equation of momentum conservation is given as follows:(8)$$\rho \frac{\partial v}{\partial t}=\rho +\nabla *\sigma$$
where stress tensor σ=−p+ϵ and ϵ is the viscous stress and *p* is pressure; hence, the momentum conservation equation becomes the following: (9)$$\rho \frac{\partial v}{\partial t}=-\nabla p+\nabla *\in +\rho$$

In the case of Newtonian fluids, the stress tensor is directly proportional to the strain rate tensor, with the proportionality constant being the fluid’s viscosity [65].

##### Energy Conservation

Energy conservation is a principle that is derived from the first rule of thermodynamics. It takes into consideration the movement and transfer of energy inside a fluid. The energy conservation equation can be expressed in a general form, as stated in reference [66]:(10)$$\rho \left(\frac{\partial E}{\partial t}+V*\nabla E\right)=-\nabla *q+\nabla *\left(\tau *V\right)+\rho g*V+Q$$

The symbol E indicates the total energy per unit mass, *q* denotes the heat flux vector, and *Q* represents any additional heat sources or sinks. The total energy comprises internal energy, kinetic energy, and potential energy. The energy conservation equation delineates the equilibrium of energy transfer resulting from convection, conduction (heat flux), stress-induced work, and external heat sources or sinks [67].

CFD models commonly use supplementary equations to represent turbulence, chemical reactions, multiphase flows, and other intricate phenomena, alongside the conservation equations. Turbulence models, such as the k-ε or k-ω models, are employed to finalise the system of equations by offering equations for the turbulent stresses and fluxes [68].

Computational fluid dynamics (CFD) has become an essential tool in multiple engineering disciplines, such as aerospace, automotive, chemical, and biomedical engineering [69]. Engineers and researchers can use it to examine intricate fluid-flow issues, enhance designs, and forecast the performance of systems in various operating situations. Due to the progress in processing resources and numerical algorithms, computational fluid dynamics (CFD) is constantly improving and addressing increasingly complex fluid dynamics issues.

### 3.5. Strengths and Limitations of Simulation Approaches

Twin-screw granulation simulation encompasses a range of computational techniques, each with distinct benefits and constraints. Table 1 presents a succinct summary of the advantages and drawbacks of the primary simulation methods examined in this analysis, which include population balance modelling (PBM), discrete element method (DEM), computational fluid dynamics (CFD), and process modelling software (PMS). The table presents the main features of each methodology, including their capacity to forecast granule characteristics, encompass particle-level interactions, replicate fluid movement, and use various modelling techniques. The text also highlights the constraints, such as the requirement for experimental verification, the computational intricacy, and the assumptions inherent in each method. The table functions as a concise guide for researchers and practitioners to assess the appropriateness of various simulation methods for their specific twin-screw granulation studies, considering factors such as the desired level of intricacy, computational resources at their disposal, and the nature of the problem being addressed. The included sources offer more information regarding the implementation and assessment of various simulation methodologies within the twin-screw granulation setting.

## 4. Simulation of Twin-Screw Granulation

Computational modelling and simulation are now indispensable for comprehending, optimising, and expanding twin-screw granulation processes. Several simulation techniques, including the discrete element method (DEM), population balance modelling (PBM), computational fluid dynamics (CFD), and coupled approaches, have been used to study the intricate relationship between material properties, process parameters, and equipment design in twin-screw granulation.

This section presents a comprehensive summary of the most advanced simulation approaches used in twin-screw granulation. It emphasises their abilities, constraints, and recent progress. The paper discusses the utilisation of discrete element method (DEM) for analysing particles at the individual level, population balance model (PBM) for forecasting the distribution and properties of granules, computational fluid dynamics (CFD) for simulating the flow and mixing of fluids, and combined methods for modelling at multiple scales.

Figure 3 presents a simulation workflow that acts as a structured framework for carrying out twin-screw granulation experiments that utilise computational approaches. The workflow provides essential stages such as problem definition, model selection, creation of geometry and mesh, determination of material properties and process parameters, numerical solution and calculation, post-processing and analysis, validation and verification, and interpretation and application. By adhering to this systematic methodology, researchers may guarantee a meticulous and replicable procedure for formulating, implementing, and evaluating twin-screw granulation simulations.

The subsequent subsections delve into the specific simulation techniques, their applications, and recent research findings, providing a comprehensive review of the current state of twin-screw granulation modelling and simulation. The section also discusses future directions and emerging trends, such as the integration of machine learning, multi-scale modelling, and sustainability considerations, which hold promise for advancing the field and optimising twin-screw granulation processes.

### 4.1. DEM Simulation of the Twin-Screw Granulation 

TSG is now recognised as a crucial technology for the uninterrupted production of solid pharmaceutical dosage forms. TSG, or continuous granulation procedures, provide several potential advantages over batch granulation processes. These include a smaller equipment footprint, enhanced process control and consistency, and greater flexibility in handling various formulations [86,87]. The incorporation of many unit activities, including powder feeding, mixing, granulation, and drying, into a single continuous process enhances the appeal of TSG as a platform for the application of quality by design (QbD) and process analytical technology (PAT) principles [87,88].

Nevertheless, achieving successful growth and expansion of TSG processes necessitates a profound comprehension of the intricate interaction of equipment design, operating circumstances, and material qualities. The screw configuration, barrel fill level, liquid addition method, and binder qualities can all have a substantial influence on the formation and growth of granules, as well as the quality attributes of the final product [9,87,89]. Obtaining a comprehensive knowledge of this process merely through experimental investigation is difficult because of the lack of transparency in the multiphase system, the brief duration of the process, and the limited opportunities for sampling and monitoring during the process.

Computational modelling tools, such as the DEM and population balance modelling (PBM), are now being used alongside experimental investigations of TSG [9,90]. These tools have proven to be highly effective in complementing experimental studies. DEM simulations offer a detailed comprehension of the particle-level mechanisms involved in the mixing, shearing, and wetting processes that control the creation and growth of granules. When combined with PBM, these models have the capability to forecast the morphology in granule size distributions and other characteristics across the TSG barrel. By combining experimental characterisation techniques, such as near-infrared (NIR) and Raman spectroscopy, with these multi-scale modelling frameworks, it becomes possible to create predictive process models. These models may then be used to inform decisions regarding equipment selection, process optimisation, and control system design.

This section of the literature review seeks to offer a thorough and detailed examination of the current innovative uses of computer modelling, specifically DEM, in order to boost understanding and improve the efficiency of TSG processes. The review will discuss the basic principles of DEM and their incorporation into multi-scale modelling frameworks, and the application of these models to analyse the impact of process parameters, equipment design, and material qualities on TSG performance. Emphasis will be placed on research that integrates modelling with experimental verification and case studies that display the application of model-driven design methods for TSG process development and optimisation. This review will also discuss emerging trends and future research directions, including the integration of particle wetting and liquid distribution phenomena, and the utilisation of high-performance computing for full-scale simulations.

#### 4.1.1. Analysis of Particle Fragmentation and Clumping in TSG

Chen et al. [91] offer useful insights into the utilisation of the Timoshenko beam bond model (TBBM) in DEM simulations to accurately forecast agglomerate breaking. The TBBM considers the axial, shear, twisting, and bending characteristics of the bonds that connect the particles. The bond parameters were calibrated using empirical measurements of the properties of the binder substance. An important benefit of the TBBM is that its bond parameters possess explicit physical significance and can be directly correlated with the characteristics of the binder material [91]. This enables a more accurate depiction of the bonding between particles in contrast to cohesive models such as JKR [59] or the parallel bond model [91].

The TBBM uses the Timoshenko beam theory to determine the forces and moments exerted on each bond. The axial force $\;\Delta F_{ax}$, shear forces $\Delta F_{ay}\;$ and $\Delta F_{az}$, twisting moment $\Delta M_{ax}$, and bending moments $\Delta M_{ay}\;$ and $\Delta M_{az}$ are calculated based on the translational and rotational displacements of the bonded particles, as described in Equations (11)–(17) in [91]. The determination of bond failure involves comparing the maximal stresses to the $\mathrm{bond}\;\mathrm{strengths}\;\mathrm{in}\;\mathrm{compression}\;\sigma_{C}$, $\mathrm{tension}\;\sigma_{T}$, and $\mathrm{shear}\;\sigma_{S}$, as outlined in Equations (25)–(27) in [91]. This approach diverges from cohesive models such as JKR, which take into account the attractive interactions between particles, and the parallel bond model [92], which assumes uniform normal and shear stiffnesses. The binding parameters were calibrated using the experimental measurements of the Young’s modulus and the tensile strength of the binder [91]. This guarantees that the simulated agglomeration behaviour accurately reflects the characteristics of the actual material.

The TBBM’s predictive capability is further demonstrated through validation versus normal impact experiments. The simulation findings offer comprehensive data on the chronological and geographical progression of the agglomerate breakage process. A significant discovery is the development of a fracture zone that has the shape of a cone and experiences intense compressive pressures. This fracture zone is encompassed by a region of tensile stress that has an arch-like shape. This provides valuable information about the distribution of stress within the agglomerate during impact, a measurement that is challenging to obtain through experimental means. The prevalence of tensile failure in bond breakage is likewise in line with experimental findings [93,94]. The primary mode of bond failure was tensile failure. The TBBM successfully replicated the accurate progression of failure modes seen in the experiment, both in terms of time and space. The TBBM has the ability to forecast agglomeration breakage by analysing the characteristics of the individual particles and binder.

The practical benefits of being able to forecast agglomeration breakup using the characteristics of the individual particles and binder are significant. It can assist in the planning and improvement of procedures related to the management and processing of agglomerates, particularly in industries such as pharmaceuticals, chemicals, and agriculture [91]. The TBBM methodology can be employed to examine the impact of distinct binder characteristics or agglomerate arrangements on the fracture behaviour, facilitating the advancement of more resilient and effective procedures. The study highlights the TBBM’s capacity as a robust instrument for examining agglomerate breaking in DEM simulations. The use of physically based binding parameters and the capability to accurately replicate realistic failure modes make it a highly promising method for predictive modelling of agglomeration behaviour.

#### 4.1.2. Influence of Fill Level on Granule and Tablet Characteristics

Matsushita, Ohsaki, Nara, Nakamura, and Watano [72] offers a thorough examination of the impact of fill level in a twin-screw granulator on the qualities of granules and tablets. This analysis employs a combination of tests and DEM simulations. The study provides useful insights into the mechanisms that determine the effect of fill level on product quality in continuous TSG.

Matsushita, Ohsaki, Nara, Nakamura, and Watano [72] conducted experimental measurements to directly assess the fill level and investigate its impact on granule size, flowability, and strength. The researchers discovered that as the amount of material in the container increased, the mean granule size also increased.

Additionally, the capacity of the granules to be compressed and their tendency to break apart decreased. These findings suggest that the flowability of the granules improved and their strength increased at higher fill levels. The researchers also quantified the hardness of tablets produced from granules collected at various fill levels and determined the interparticle adhesion force using the Rumpf equation. The Rumpf equation, often referred to as the Rumpf model or Rumpf’s hypothesis, is a theoretical formula that establishes a relationship between the tensile strength of a granular substance and its porosity, as well as the forces of adhesion between particles. The proposal was initially put forward by Rumpf [95]. The Rumpf equation is expressed as follows:(11)$$\sigma_{z}=\frac{1-\varepsilon }{\varepsilon }\;*\;k\;*\frac{H}{d_{p}^{2}}$$

$\sigma_{z}\;$represents the tensile strength of the granular material, ε represents the porosity of the granular material, k represents the coordination number (which is the average number of connections per particle), *H* represents the interparticle adhesion force per contact point, and $d_{p}$ represents the diameter of the particles.

In Matsushita et al.’s study, the authors utilise an empirical correlation between porosity (*ε*) and coordination number (*k*) proposed by Ridgeway [96] to estimate the coordination number for application in the Rumpf equation. The equation is given by
(12)$$\varepsilon =1.072-0.1193k+0.00431k^{2}.$$

The symbol *ε* represents the porosity of the granular material, whereas k represents the coordination number. The coordination number denotes the mean number of interactions per particle in a granular system.

In order to attain more insight into the processes involved, Matsushita, Ohsaki, Nara, Nakamura, and Watano [72] utilised the DEM simulations to analyse the movement of particles in the twin-screw granulator. The DEM model employed in this investigation was founded on the Johnson–Kendall–Roberts (JKR) theory, which describes the adhesion forces between particles. Additionally, the Hertz–Mindlin contact model was utilised to account for both normal and tangential forces [72] The simulations were conducted utilising the EDEM 2017 commercial program, and the model geometry was constructed utilising real-scale computer-aided design data of the twin-screw granulator [72]. The DEM simulations demonstrated concordance with experimental results on the mixing diffusion coefficient at various screw speeds, hence confirming the model’s capability to accurately depict particle movement within the granulator [72]. The simulations revealed that when the screw speed increased, both the average residence time and fill level reduced. The study by Matsushita, Ohsaki, Nara, Nakamura, and Watano [72] revealed that the compressive force exerted on particles was shown to rise as the fill levels increased, while it remained consistent at a constant fill level, irrespective of the screw speed. This finding is significant.

Matsushita, Ohsaki, Nara, Nakamura, and Watano [72] suggest a mechanism in which the high fill level in the twin-screw granulator enhances the compressive force on particles, resulting in increased granule strength. This proposal is based on the combination of experimental and modelling findings. Consequently, the use of granules from high fill level conditions leads to tablets that have increased interparticle adhesion forces and hardness [72].

The work demonstrates the efficacy of integrating experiments with DEM simulations to clarify the intricate mechanisms in granulation processes. The utilised DEM technique, which integrates the JKR adhesion model and Hertz–Mindlin contact model, facilitated a comprehensive examination of forces at the particle level and their reliance on process parameters such as fill level and screw speed. The generated insights can assist in the logical development and improvement of TSG operations.

Overall, Matsushita et al.’s paper enhanced comprehension of fill level effects in TSG through the utilisation of a comprehensive experimental and DEM simulation methodology. The study emphasises the significance of fill level as a crucial process parameter and offers detailed insights that might assist in optimising the process.

#### 4.1.3. Tracking and Mixing of Materials in Twin-Screw Feeders 

Toson and Khinast [97] also conducted a thorough investigation on refill techniques for a twin-screw feeder using a DEM model in their research. The authors emphasise the significance of considering the blending of previous and current material batches in the feeder to ensure precise material tracking in continuous manufacturing lines. The study relies on a previously published open-access dataset that documents the complete discharge process of a twin-screw feeder [98]. DEM simulation examined the residence time distribution (RTD) and mixing behaviour of particles within a twin-screw feeder during a whole discharge operation.

The results obtained from the DEM analysis show intricate flow patterns within the feeder. The analysis indicates that material is discharged early from the region positioned above the agitator, whereas the material surrounding the rotational axis of the agitator tends to become stuck and is only discharged when the fill levels are lower. This observation was made by Toson and Khinast in 2023. In order to forecast the behaviour during multiple refill events, the authors introduce three models that become progressively more complex: (1) a basic exponential residence time distribution (RTD) that assumes flawless mixing of material batches, (2) an RTD model derived from DEM results, and (3) particle-level material tracking achieved by extrapolating the DEM results using a “relay race” approach [99].

The initial model, known as the exponential survival function, is characterised by the $\mathrm{time}\;t_{r}$ and the remaining $\mathrm{fill}\;\mathrm{level}\;m_{r}$ of the feeder refills [97]. The second model uses particle-based residence periods derived from the DEM data to iteratively compute survival functions across many refills. The third model, known as the “relay race,” is a strategy in DEM extrapolation that accurately monitors the positions of particles during refill occurrences in order to forecast the washout behaviour [99]. The authors conducted a comparison of the performance of these models at various refill levels and determined that the simple perfect mixing model is a satisfactory approximation for refills at or below 30% fill level. However, for higher refill levels, the RTD and relay race models provide more accurate results [97].

The DEM results unveiled intricate flow patterns within the feeder. The powder bed in the hopper was partitioned into layers with a thickness of 2 cm for the purpose of conducting quantitative analysis. The discharge curves of different layers indicated that the material positioned above the agitator axis exhibited early discharge, whereas the material surrounding the rotating axis of the agitator became trapped and only discharged at lower fill levels. The researchers, Toson and Khinast (2023) [62], found that the behaviour described was consistent across different initial fill levels. They discovered that the discharge of all material layers was only uniform when the fill level was at or below the agitator axis. The residence time distribution (RTD) is a fundamental term used to describe the mixing and flow characteristics of particles in processing equipment. Within the twin-screw feeder framework, the residence time distribution (RTD) characterises the duration that particles remain inside the feeder prior to being released. The DEM simulation yielded useful insights into the resident time distribution (RTD) by monitoring the duration that individual particles remained within the system. The authors computed the survival function, denoted as S(t), which is the inverse of the cumulative distribution function, denoted as F(t), for the residence times [62].

The DEM simulation and the subsequent residence time distribution (RTD) data served as a basis for comprehending the intricate mixing and flow dynamics within the twin-screw feeder. The findings obtained from the DEM investigation facilitated the creation of simplified models that can forecast the movement of materials during numerous feeder refills. This is essential for ensuring precise material monitoring in continuous manufacturing lines [62].

The study highlights the significance of considering the blending of material batches within the feeder to ensure precise material tracking in continuous manufacturing lines. The suggested reduced-order models, including the RTD and relay race models, can be customised to suit individual equipment and materials, offering a helpful tool for forecasting the behaviour during successive feeder refills. The authors propose that future research might involve the comparison of various feeder designs, operating circumstances, and material properties using the established tool chain of DEM modelling combined with residence time distribution (RTD) modelling and the relay race extrapolation technique [97].

#### 4.1.4. Powder Blending in Twin-Screw Granulators

Mateo-Ortiz, Villanueva-Lopez, Muddu, and Doddridge [87] examined the practicability of blending dry powders in the pre-melting regions of a twin-screw extruder through the utilisation of DEM modelling and near-infrared (NIR) spectroscopy [87]. The authors’ objective was to integrate the blending and extrusion/granulation processes into a single apparatus, thereby facilitating continuous manufacturing and enhancing process efficiency. The study used DEM simulations with the Hertz–Mindlin contact model to obtain insights into the mixing patterns within the extruder at the particle level. This information was then used to assist in the selection of the screw configuration prior to conducting experiments [87]. The angle of repose test was used to calibrate the parameters of the DEM model. The simulations demonstrated satisfactory axial mixing for all screw configurations examined; however, satisfactory radial mixing was only observed in the design featuring 90-degree kneading elements [87].

The study used a Coperion ZSK-18 extruder equipped with NIR spectroscopy to assess the homogeneity of blends. This was achieved by investigating the screw arrangement, screw speed, and powder feed rate [1]. The calibration models for NIR were created using projection-to-latent structures (PLS) regression. The equation used to estimate the mass of the sample scanned by the NIR probe is as follows.

The equation for *M* is given by
(13)$$M=\rho \left[\pi {\left(\frac{d}{2}\right)}^{2}+d\left(t_{acq}\;*\;V_{pow}\right)\right]*H$$

The variables in the equation are defined as follows: ρ represents the bulk density of the sample, d represents the diameter of the NIR beam, $t_{acq}$ represents the acquisition time, $V_{pow}$ represents the linear velocity of the powders, and H represents the experimental depth of penetration of the NIR beam [87]. The findings indicate that the twin-screw extruder is capable of effectively blending two powder feed streams at a unit dose scale for the examined formulation containing a cohesive API [87]. The residence time distribution (RTD) profiles of screw topologies consisting solely of conveying components exhibit similarities to a plug flow reactor (PFR), whilst configurations containing kneading devices display a greater resemblance to an ideal continuous stirred tank reactor (CSTR) [87]. Mateo-Ortiz, Villanueva-Lopez, Muddu, and Doddridge [87] suggest that these investigations can aid in the advancement of efficient hot melt and wet extrusion/granulation techniques using twin-screw extruders for the continuous production of oral solid dose products.

#### 4.1.5. Particle Breakage in Twin-Screw Pulping

Cheng, Gong, Zhao, Zhang, Lv, and Ren [75] examined the impact of twin-screw pulping technology on the performance of straw pulping. Cheng, Gong, Zhao, Zhang, Lv, and Ren [75] utilised the DEM and experimentally validated their findings using the Tavares mathematical model. The researchers utilised SolidWorks to develop a 3D representation of the twin-screw pulping machine. Additionally, they employed the Tavares breaking model to construct a DEM model for the fragmentation of straw particles. This model assumes that the likelihood of fragmentation follows an upper truncated log-normal distribution. Additionally, it considers that the breakage ratio energy drops because of cumulative damage during particle impact. The Voronoi fracture equation is used to determine the shape of the fragment, which is then distributed based on the Gaudin–Schumann function [75]. The expression for the chance of breakage is as follows: The probability of event e, denoted as $P_{o}\left(e\right),\;$is equal to one-half. The expression $\left[1+\left(\frac{lne^{*}-lne_{50}}{\sqrt{2\delta^{2}}}\right)\right]\;$is evaluated. The symbol $e^{*}$ represents the relative breakage energy, $e_{50}$ represents the average breakage energy, and $\delta^{2}$ represents the variance of the log-normal distribution. The calculation of the relative breakage energy is as follows: The equation (Equation (6)) represents the relationship between $e^{*},$ $e_{max}\;$, *e*_$e_{max}$, and e, where $e^{*}$ is calculated as the ratio of *e_max* to the difference between $e_{max}e$ and e. The average breakage energy is determined by the equation $e_{50}=e_{\infty }$ multiplied by the quantity of 1 plus the ratio of $d_{0}$ to L raised to the power of φ. This is given in Equation (7) by [75]. 

The simulations were performed using different combinations and configurations of twin-screw spiral casings and tooth groove angles. The casing combination of negative–positive–negative–positive (NPNP) and the tooth groove angle arrangement of 45°-30°-15° resulted in the greatest number of shattered straw particles [75]. A Box–Behnken experimental design was employed to construct a mathematical model that correlates pulp yield with the tooth groove angle (A), screw speed (B), and straw moisture content (C). The resultant quadratic polynomial equation is given as Equation (10) in [75], as follows:(14)$$\theta =90.2+0.93A+3.20B+1.94C-0.10AB-0.03AC+1.8BC-7.86A^{2}-16.39B^{2}-7.21C^{2}$$

The results of the multi-factor trials indicated that the screw speed had the most significant impact on pulp yield, followed by the moisture content of the straw and the angle of the tooth groove. The study determined that the most favourable parameters for achieving a pulp yield of 92.5% were a screw speed of 550 r/min, a tooth groove angle of 30°, and a straw moisture content of 65%.

The validation studies conducted on a twin-screw pulping machine have proven that the combination of an NPNP spiral casing with a tooth groove angle arrangement of 45°-30°-15° yielded the most optimal pulping performance [75]. The moisture content of straw had a substantial impact on the pulping process, as a moisture level of 60–70% resulted in successful lignin degradation and pulping. Decreased or increased moisture content negatively impacted the efficiency of the pulping process. The straw particles were effectively broken down without affecting the activity of lignin-degrading enzymes while using screw speeds ranging from 500 to 550 revolutions per minute (r/min). The practical results corroborated the simulation predictions, confirming the suitability of using DEM and the Tavares model for analysing straw breakage in twin-screw pulping [75].

This work highlights the effective utilisation of DEM and the Tavares breakage model to simulate and optimise the process of straw pulping in a twin-screw pulping machine. By utilising both simulation and experimental validation, one can gain useful insights into how to improve the performance and yield of straw pulping. This is achieved by adjusting the design of the equipment and the operating parameters. The Tavares model is a reliable and precise method for representing the intricate process of particle fragmentation in twin-screw pulping systems. The crucial equations derived in this study, including the breakage probability expression, calculations of relative and average breakage energy, and the quadratic polynomial equation that links pulp yield to process parameters, establish a basis for future research and optimisation of twin-screw pulping technology [75].

#### 4.1.6. Mixing Dynamics of Cohesive Particles in Twin-Screw Mixers

Karkala and Ramachandran [74] devised an extensive DEM model to examine the blending dynamics of adhesive particles in a co-rotating twin-screw mixer (TSM). The Johnson–Kendall–Roberts (JKR) contact model was utilised to simulate the cohesive forces between particles. The JKR model’s fundamental equations establish a connection between the normal contact force (${\vec{F}}_{\vec{N}}$) and other parameters such as the equivalent particle radius ($R_{eq}$), equivalent Young’s modulus ($E_{eq}$), contact radius (*a*), and surface energy (γ) [74].
(15)$${\vec{F}}_{\vec{N}}=\frac{4}{3}\;*\frac{E_{eq}}{R_{eq}}*\;a^{3}-4*{\left(\pi E_{eq}\gamma \right)}^{\frac{1}{2}}\;*\;a^{\frac{3}{2}}$$

The normal overlap ($\delta_{T}$) is given as follows:(16)$$\;\delta_{N}=\frac{a^{2}}{R_{\mathrm{eq}}}-\sqrt{\frac{4a\pi \gamma }{E_{eq}}}{*a}^{3}$$

Tangential forces (${\vec{F}}_{\vec{T}}$) are determined using the Mindlin–Deresiewicz model, which establishes a relationship between the tangential overlap ($\delta_{T}$) and the equivalent shear modulus ($G_{eq}$) [74].
(17)$${\vec{F}}_{\vec{T}}=-\left(8G_{eq}\;\sqrt{R_{eq}\delta_{N}}\right)*\delta_{T}$$

The magnitude of these tangential forces is constrained by Coulomb’s law, which incorporates the coefficient of static friction ($\mu_{S}$) [100]:(18)$${\vec{F}}_{\vec{T}}=sign({\vec{F}}_{\vec{T}})*min\left(|{\vec{F}}_{\vec{T}}|,|\mu_{S}{\vec{F}}_{\vec{N}}\;|\right)$$

The researchers adjusted the DEM particle parameters by conducting angle of repose simulations for particles with good flow (GF) and poor flow (PF), and dynamic yield strength simulations for particles with very poor flow (VVPF) [74]. The meticulous calibration method guaranteed that the model precisely depicted the cohesive behaviour of the particles. The study conducted a simulation of the mixing process of a binary system consisting of P1 and P2 particles in a mass ratio of 80:20. The total throughput of the system was 3 kg/h. The attributes of P1 were maintained at a constant level, while the size, density, and flowability of P2 were altered at three distinct levels. The flowability of P2 was classified according to Carr’s categorisation, ranging from GF to VVPF [74]. The TSM model was partitioned into five equidistant compartments (C1–C5) to examine holdup, residence time, and degree of mixing. The accuracy of the model was verified by comparing the anticipated average residence times with the values obtained from experiments [101,102].

The simulations yielded useful insights into the mixing mechanisms of particles with varying flowabilities. Mixing mostly took place in three specific areas of the conveying sections for GF and PF particles: on a single screw, between the screws at the top of the TSM, and between the screws at the bottom of the TSM. The mixing was facilitated by the repetitive cycle of agglomeration fragmentation and recombination during the process of rolling transportation. On the other hand, VVPF particles formed smaller clusters and were evenly spread across both screws because of their strong adherence with the walls [74]. Having a clear comprehension of the mixing zones and mechanisms is essential for maximising the efficiency and effectiveness of TSM design and operation. The particle flowability had a significant impact on both the steady-state holdup and mean residence duration, in comparison to the effects of size and density. The VVPF particles demonstrated a holdup that was 25–50% greater and had longer mixing periods in comparison to the GF particles. The quantification of mixing efficiency was performed by calculating the relative standard deviation (RSD) of P2 composition across the TSM compartments [74].

The VVPF particles had the highest mixing rate, whereas the GF particles demonstrated the slowest mixing rate and the most pronounced demixing. Introducing kneading elements in C4 resulted in enhanced mixing for the GF and PF particles, whereas it had minimal impact on VVPF mixing. It is suggested adjusting screw designs to enhance mixing efficiency based on the results obtained. Specifically, they recommended solely employing conveying elements for VVPF particles, expanding the kneading sections for PF particles, and positioning the kneading elements closer to the TSM output for GF particles. These modifications aim to minimise demixing that may occur afterwards [74]. These recommendations emphasise the usefulness of the model in directing the design of TSM for achieving cohesive particle mixing.

#### 4.1.7. Effect of Particle Shape on Conveying Properties in TSG

Zheng, Govender, Zhang, and Wu [70] thoroughly examine how particle form affects the conveying properties in a full-scale twin-screw granulator. Zheng, Govender, Zhang, and Wu [70] employed a GPU-enhanced DEM approach to conduct their inquiry. The study examines multiple particle shapes, such as sphere, cube, bilunabirotunda (Biluna), and hexagonal prism (HexP), to model the twin-screw granulation (TSG) process. TSG is a vital continuous production method utilised in various sectors [2,70].

Govender, Wilke, and Kok [56] utilise the Blaze DEM-GPU algorithm, which employs an explicit forward Eulerian time integration approach on GPU to solve the motion equations of particles. Newton’s second law describes the motion of particles in terms of both translation and rotation.
(19)$$m_{i}\frac{dV_{i}}{dt}=F_{C,i}+m_{i}g$$
(20)$$I_{i}\frac{d\left(\omega_{i}\right)}{dt}=T_{i}$$

$\;m_{i}\;$represents the mass of the particle, $I_{i}\;$represents the moment of inertia, $V_{i}$ represents the translational speed, g represents the gravitational acceleration, $\omega_{i}$ represents the angular speed, $F_{C,I}$ represents the total contact force, and $T_{i}$ represents the torque [70]. The contact forces between particles are determined using the precise volume contact detection method, which is more accurate than previous algorithms that rely on distance-based force resolution [70]. The normal force is determined by applying the Kelvin–Voigt linear viscoelastic spring-dashpot model.
(21)$$F_{n}=\left(K_{n}\Delta V^{\frac{1}{3}}\right)n-C_{n}\left(V_{R}*\;n\right)n$$

$K_{n}$ represents the stiffness of the spring, n refers to the normal direction of the force, $C_{n}$ represents the damping coefficient, and $V_{R}$ represents the relative velocity between two particles in contact [70]. The tangential force is calculated using the Cundall–Stark model:(22)$$F_{T}=-K_{T}\left(V_{T}dt\right)-C_{T}V_{T}+F{'}_{T}$$

$F{'}_{T}$ represents the component of the force that is parallel to the current plane. $K_{T}$ refers to the stiffness of the spring in the same direction. $C_{T}$ represents the damping coefficient in the same direction. $V_{T}$ represents the relative velocity in the same direction [70]. The study examines the particle flow patterns, residence time distribution, and power consumption of various particle shapes in the TSG process. According to the findings, spherical particles have the highest transit speed and the lowest amount of time spent in a certain place. On the other hand, polyhedral particles have a more intricate movement pattern and spend more time in a specific location because of the complex collisions among particles and between particles and walls [70]. The characterisation of the resident time distribution is accomplished by utilising the average residence time.
(23)$$t_{m}=\overset{\;\infty \;}{\underset{0}{\int }}\;tE\left(t\right)dt\;$$

The residence time variance is calculated as follows:(24)$$\sigma^{2}=\sum_{i=0}^{\infty }{\left(t_{i}-t_{m}\right)}^{2}E\left(t_{i}\right)$$
and the function that represents the distribution of ages at which individuals exit cumulatively is calculated as follows:(25)$$F\left(t\right)=\overset{\;t\;}{\underset{0}{\int }}\;E\left(t\right)dt$$

The RTD function is denoted as *E*(*t*), and the mean residence time is represented by $t_{m}$ [70].

The power consumption is calculated by adding together the energy wasted in both the normal and tangent directions of each contact [70,103]. The numerical results indicate that particle form has a substantial impact on the conveying properties during the TSG process. According to [70], spherical particles have a flow pattern that is more similar to ideal plug flow, whereas cubic particles have a flow pattern that is more similar to perfect mixing flow. According to the study, polyhedral particles require more power throughout the TSG process because they collide with the wall more frequently and with more force [70].

The utilisation of the GPU-enhanced DEM methodology facilitates a comprehensive comprehension of the influence of particle morphology on the conveying properties within a full-scale twin-screw granulator. The collected information can facilitate a deeper comprehension of the TSG process at the particle level and enable the optimisation of process parameters to enhance granulation performance [70]. The study emphasises the significance of considering the shape of particles when conducting numerical simulations of the TSG process. It also showcases the effectiveness of the GPU-enhanced DEM technique in simulating particles with intricate shapes in full-scale process simulations [70].

The utilisation of computational modelling methods, specifically DEM, has enhanced our comprehension of the intricate dynamics that drive TSG processes. DEM simulations at the particle scale have yielded useful insights into the behaviour of powders in TSG, specifically in terms of mixing, shearing, and wetting. Additionally, the use of multi-scale DEM frameworks has allowed for the prediction of granule size distributions and other quality aspects. The incorporation of these models with experimental characterisation techniques, including NIR and Raman spectroscopy, has enabled the creation of prognostic process models that can assist in the selection of equipment, optimisation of processes, and design of control systems.

The reviewed literature shows that DEM models have effectively been used to study the impact of different process parameters, including screw configuration, barrel fill level, the liquid addition method, and binder characteristics, on TSG performance. These models have also been employed for the purpose of comparing various granulation processes, such as wet and dry binder addition, and for the optimisation of energy usage and scale-up. The integration of particle wetting and liquid distribution phenomena into DEM models has significantly improved their ability to make accurate predictions. This advancement allows for the simulation of more intricate compositions and process conditions.

Nevertheless, there are substantial prospects for further investigation and advancement in this domain. By combining DEM with other modelling tools like CFD and finite element analysis (FEA), it becomes possible to simulate process conditions with more realism. This includes studying the impact of barrel temperature and pressure on granule qualities. Utilising high-performance computing platforms, such as systems based on GPUs, could enable the modelling of complete TSG equipment, offering significant insights into the scaling up and optimisation of processes. Moreover, the utilisation of these modelling techniques in other sectors, such as pulp and biomass processing, has the potential to foster the creation of manufacturing processes that are both more energy-efficient and sustainable.

Overall, the utilisation of computational modelling, namely DEM, has significantly transformed our comprehension of TSG procedures. The utilisation of these technologies has facilitated the creation of prognostic process models that may direct the planning, enhancement, and management of TSG activities, resulting in enhanced product excellence, decreased time and cost for development, and heightened process adaptability. As the pharmaceutical sector increasingly adopts continuous production and quality by design (QbD) concepts, it will become crucial to incorporate these modelling methodologies into the product development and manufacturing workflow. In the future, we can anticipate further progress in TSG modelling due to advancements in computer power, experimental characterisation methodologies, and modelling frameworks.

### 4.2. Population Balance Modelling of the Twin-Screw Granulation Process 

Population balance modelling (PBM) is a highly effective mechanistic modelling technique for particle processes, particularly twin-screw wet granulation. Particle-based models (PBMs) allow for the monitoring of changes in particle properties, such as size, liquid content, and porosity, as they are influenced by different rate processes, including aggregation, breakage, consolidation, and liquid addition [104,105,106]. PBMs are ideal for simulating twin-screw wet granulation, an intricate process that encompasses various competing mechanisms in the distinct zones of the twin-screw granulator (TSG). Being able to forecast these characteristics is essential for comprehending and enhancing the TSG procedure, as they have a direct influence on the ultimate granule quality.

The population balance equation (PBE) for a particulate system undergoing aggregation and breakdown in a TSG can be expressed in the following generic form [1,4]: The selection of model dimensionality is a crucial option while designing a PBM for a TSG. One-dimensional (1D) PBMs, which primarily monitor particle size, are computationally efficient but may not fully represent the intricate nature of the granulation process in a TSG [104,106]. Barrasso, El Hagrasy, Litster, and Ramachandran [104] and Ismail, et al. [107] created two-dimensional population balance models (PBMs) that can track both the size and liquid content of particles. This approach offers a more thorough description, but it requires more computer resources. The selection of dimensionality relies on the particular objectives of the model and the computational resources that are accessible. Higher-dimensional models offer greater depth of understanding of the TSG process but may pose greater difficulties in terms of solving and calibrating. The lumped-parameter technique provides a solution by reducing multi-dimensional PBMs to interconnected 1D equations, while also determining particle attributes depending on size [105]. This methodology, employed by Barrasso, El Hagrasy, Litster, and Ramachandran [104] and Shirazian, et al. [108] in the context of TSG modelling, merges the rapid computational capabilities of 1D PBMs with the capacity to forecast numerous characteristics. Shirazian, Darwish, Kuhs, Croker, and Walker [108] employed this method to forecast the distributions of both size and liquid content in a TSG.
(26)$$\frac{d\left[n\left(v,l,x,t\right)l\left(v,x,t\right)\right]}{\partial t}+\frac{\partial \left(V_{x}\;n\;l\right)}{\partial x}=B_{l}\left(v,l,x,t\right)-D_{l}\left(v,\;l,\;x,\;t\right)$$

In this context, l represents the liquid content, $V_{x}$ denotes the velocity along the length of the TSG, and $B_{l}$ and $D_{l}$ refer to the birth and death terms for liquid content. Another crucial factor to consider is the variation in spatial patterns within the TSG. To solve this issue, compartmental PBMs employ a strategy of separating the TSG into several spatial zones or compartments, and then applying PBM separately in each of these zones [104,105,106,108]. Van Hauwermeiren, Verstraeten, Doshi, am Ende, Turnbull, Lee, De Beer, and Nopens [105] devised a two-stage compartmental PBM for a TSG, where the wetting and kneading regions were modelled independently. This technique enables the utilisation of diverse kernels and parameters in each zone, thereby accurately representing the distinct prevailing mechanisms. Furthermore, it allows for the anticipation of granule characteristics at various positions throughout the TSG, which can be quite advantageous for comprehending and improving the process.

Regime-separated PBMs undergo an additional step by assuming distinct flow conditions in certain spatial zones of the TSG. Shirazian, Darwish, Kuhs, Croker, and Walker [108] hypothesised that the transporting portions of a TSG exhibit plug flow, while the kneading elements exhibit mixed flow. These methods enable a more precise depiction of the diverse circumstances inherent in the TSG. The selection of flow regime has a substantial influence on the model’s forecasts since it dictates the distribution of residence time and the level of mixing in each zone. A precise depiction of the flow conditions is essential for accurate simulation of the TSG process. Experimental data are essential in the development and validation of process-based models (PBMs) for TSG. The collection of data that provide information about the distribution of granule size, liquid, and porosity throughout the length of the TSG is highly important [6,104,106,108]. Van Hauwermeiren, Verstraeten, Doshi, am Ende, Turnbull, Lee, De Beer, and Nopens [105] and Verstraeten, Van Hauwermeiren, Lee, Turnbull, Wilsdon, am Ende, Doshi, Vervaet, Brouckaert, Mortier, Nopens, and Beer [15] gathered these data for a TSG, allowing for thorough model construction and validation. The presence of such data enables the adjustment of model parameters and the evaluation of model precision at various points within the TSG.

Additionally, it offers valuable information regarding the development of granule characteristics and the prevailing processes within each TSG zone. The conventional approach for modelling TSG using a PBM solution involves the use of discretisation techniques. Barrasso, El Hagrasy, Litster and Ramachandran [104] and Shirazian, Darwish, Kuhs, Croker, and Walker [108] utilised the cell average approach, whereas Van Hauwermeiren, Verstraeten, Doshi, am Ende, Turnbull, Lee, De Beer, and Nopens [105] implemented the finite volume scheme. Parameter estimation involves adjusting model predictions to align with experimental TSG data [105,107]. The selection of the discretisation method can have a significant impact on both the accuracy and efficiency of the solution. Finite volume approaches have demonstrated excellent conservation qualities and the ability to manage intricate kernels in TSG modelling [105]. Parameter estimation is a crucial stage in model building since it impacts the model’s capacity to accurately replicate experimental observations from the TSG. Aggregation and breakage kernels play a crucial role in population balance models (PBMs) for TSG, as they dictate the rates at which these processes occur. Barrasso, El Hagrasy, Litster, and Ramachandran [104] employed the Madec kernel to aggregate in a TSG model, which takes into consideration both the size and liquid content.
(27)$$\;\beta \left(s,\;s^{\prime },z,\;t\right)=\beta_{0}\;\left(V+V^{\prime }\right){\left[{\left(\frac{LC+LC^{\prime }}{2}\right)}^{\alpha }\left(1-\frac{LC+LC^{\prime }}{2}\right)\right]}^{\delta },$$
where $\beta_{0}$, *α*, and *δ* are parameters, and *LC* and *LC′* represent the liquid contents of the aggregating particles. The user is referring to a rate coefficient, denoted as *β*_0_, and two exponents that quantify the extent of dependency on liquid, represented by *α* and *δ*. The parameter *δ* quantifies the manner in which the aggregation rate achieves its highest value at an appropriate liquid concentration. Van Hauwermeiren, Verstraeten, Doshi, am Ende, Turnbull, Lee, De Beer, and Nopens [105] created a new kernel to represent the bimodal granule size distributions (GSDs) that are seen in the wetting zone of a TSG.
(28)$$\beta \left(x,\;\varepsilon \right)=\left(\frac{top_{1}}{2}\;\left[1+tanh\left(\frac{R_{1}^{3}-{\left(x^{2}+\varepsilon^{2}\right)}^{\frac{1}{2}}}{d1}\right)\right]-\frac{top_{1}-top_{2}}{2}\left[1+tanh\left(\frac{R_{2}^{3}-{\left(x^{2}+\varepsilon^{2}\right)}^{\frac{1}{2}}}{d2}\right)\right]\right)*\;\left(x^{\frac{1}{3}}\;\varepsilon^{\frac{1}{3}}\right)$$

This kernel employs a step function to create a distribution with two distinct modes. The parameters of the step function determine the position and sharpness of the step. The selection of kernel functions is determined by a comprehensive study of the granulation mechanisms in the TSG and the observed experimental patterns. The liquid-to-solid (L/S) ratio is universally recognised as the primary factor influencing the granule size distribution (GSD) in TSG. Higher L/S ratios result in bigger granule sizes [105,108]. Shirazian, Darwish, Kuhs, Croker, and Walker [108] and Van Hauwermeiren, Verstraeten, Doshi, am Ende, Turnbull, Lee, De Beer, and Nopens [105] both emphasise this discovery regarding TSG. The L/S ratio quantifies the quantity of liquid present for granulation in the TSG, hence influencing the degree of aggregation. When the liquid-to-solid (L/S) ratio is low, there may not be enough liquid to completely form granules from the powder, resulting in a higher proportion of tiny particles. At high liquid-to-solid ratios, there is a greater likelihood of widespread aggregation taking place, which causes a change in the granule size distribution towards bigger sizes. When the liquid-to-solid (L/S) ratios are low, two or more peaks in the grain size distribution (GSD) are commonly seen in TSG. This is caused by the uneven distribution of liquid. Van Hauwermeiren, Verstraeten, Doshi, am Ende, Turnbull, Lee, De Beer, and Nopens [105] created a kernel that was specifically designed to represent this behaviour in a TSG. The uneven dispersion of fluids can result in certain areas of the powder bed being excessively moistened while others are insufficiently moistened. This can lead to the formation of a combination of large, strongly clustered particles and smaller, less-clustered particles, resulting in a distribution of particle sizes that has two or more distinct peaks, known as a bimodal or multimodal granule size distribution (GSD). It is crucial to capture this behaviour to accurately anticipate the complete GSD in a TSG.

The studies offer valuable insights into the prevailing rate mechanisms in various TSG zones. Shirazian, Darwish, Kuhs, Croker, and Walker [108] discovered that aggregation is the primary mechanism for transporting elements, whereas breakage plays a substantial role in the kneading elements of a TSG. This aligns with the anticipated mechanisms in these areas. When transporting elements, the powder is moved without much blending or deformation, which enables the particles to come together and form aggregates. During the process of kneading, the application of intense shear forces might result in the fragmentation of the granules. Comprehending these mechanisms that are distinct to each zone is essential for optimising processes in a targeted manner in a TSG.

Research has demonstrated that compartmental and regime-separated PBMs are more effective than single-compartment models in accurately representing the diverse GSDs for TSG [105,108]. Van Hauwermeiren, Verstraeten, Doshi, am Ende, Turnbull, Lee, De Beer, and Nopens [105] and Shirazian, Darwish, Kuhs, Croker, and Walker [108] provide evidence of the efficacy of these methods in TSG modelling. By taking into consideration the distinct conditions and mechanisms present in each TSG zone, these models can offer more precise forecasts of the complete GSD. On the other hand, single-compartment models may have difficulty representing the complete intricacy of the TSG process, particularly when there are large variations in the characteristics of the granules along the length of the TSG. Barrasso, El Hagrasy, Litster, and Ramachandran [104] expanded their TSG model to incorporate numerous solid constituents, which represent the API and excipient. A composition-dependent aggregation kernel was implemented to model the attractive or repulsive interactions between components.
(29)$$\psi \left(x,x^{\prime }\right)=\;\exp\left[-a_{AB}\left(\;x+x^{\prime }-2xx^{\prime }\right)\right]$$

The parameter $-a_{AB}$ determines the strength of the interaction, while *x* and *x’* represent the compositions of the aggregating particles. This update enables the forecasting of granule composition in a TSG, which is crucial for ensuring consistent distribution of API in the product. Notwithstanding these progressions, there are still obstacles in creating a comprehensive and thoroughly verified PBM for TSG. A profound comprehension of the fundamental physical mechanisms, a wealth of experimental data obtained from the TSG, and reliable numerical approaches are necessary. The discussion so far, whilst offering useful insights, also emphasises the necessity for additional research in this field. Major obstacles involve the creation of physically grounded kernels for TSG, the collection of comprehensive experimental data for calibration and validation, and the effective resolution of multi-dimensional and multi-component models.

Van Hauwermeiren, Verstraeten, Doshi, am Ende, Turnbull, Lee, De Beer, and Nopens [105] emphasise the importance of enhancing kernels and the possibility of connecting model parameters to material qualities and process circumstances in TSG. This has the potential to result in more accurate predictive models and decrease the necessity for calibration studies. If it were possible to forecast model parameters using quantifiable material properties (such as powder size distribution and binder viscosity) and process settings (such as screw speed and L/S ratio), the model might be utilised to make predictions about the granule properties in a TSG beforehand. This would significantly improve the usefulness of the model for TSG process design and optimisation. In Shirazian et.al. (2017) [108], the authors highlight the significance of comprehending the prevailing mechanisms in every TSG zone. Their technique, which involves separating the regime, offers a framework for this purpose. However, additional research is required to thoroughly analyse and understand these mechanisms. One approach to investigating the micro-scale particle interactions in each TSG zone is to conduct detailed experimental research on the TSG, potentially in combination with other modelling techniques such as DEM simulation. Such a comprehension could direct the advancement of kernels that are better grounded in physical principles and establish a foundation for forecasting kernel parameters for TSG. Ismail, Shirazian, Singh, Whitaker, Albadarin, and Walker [6] observe that multi-dimensional PBMs have the capability to forecast supplementary characteristics of granules, such as porosity in TSG. Nevertheless, they also emphasise the computational difficulties and the necessity for appropriate multi-dimensional kernels. A comprehensive comprehension of how the process circumstances and material qualities of TSG influence the development of these supplementary properties is necessary for the creation of such kernels. Conducting experimental research on TSG, with a specific focus on these features, and utilising modern characterisation techniques such as X-ray tomography, have the potential to offer valuable insights. Kumar, Peglow, Warnecke, Heinrich, and Mörl [106] examine the incorporation of PBMs with other modelling methodologies, such as DEM, for TSG modelling. Hybrid models have the potential to offer a comprehensive depiction of the TSG granulation process, but they also bring forth more intricacy. The discrete element method (DEM) can offer precise insights into the interactions between particles and the patterns of flow in a TSG. This knowledge can be used to enhance the accuracy of the kernels and assumptions about flow regimes in the PBM. Nevertheless, the expense of performing DEM calculations and the difficulties of integrating them with PBM must be resolved in order for this method to be viable for TSG modelling.

Overall, the reviewed experiments clearly illustrate the effectiveness and promise of PBMs in simulating twin-screw wet granulation. They offer useful observations into the fundamental mechanisms and factors that affect the characteristics of granules in a TSG. Nevertheless, they also emphasise the persistent obstacles and the necessity for additional investigation to create completely anticipatory TSG models. Due to ongoing progress in experimental methods, computational techniques, and mechanistic understanding, PBMs have become essential in the development and optimisation of the TSG process, a significant pharmaceutical production process. The primary objective is to create models that can accurately forecast granule characteristics using formulation and process inputs, facilitating the creation of resilient and effective TSG processes. To accomplish this objective, it is imperative to establish a strong partnership between researchers conducting experiments and those developing models. Additionally, a continuous and dedicated endeavour is necessary to enhance our comprehension of the intricate phenomena associated with twin-screw wet granulation.

### 4.3. Coupled Simulation of Twin-Screw Granulation

Twin-screw wet granulation (TSG) has emerged as a viable continuous processing approach in the pursuit of increasing quality by design (QbD) principles in pharmaceutical manufacture. Nevertheless, comprehending and enhancing TSG procedures is hindered by the intricate interaction of material properties, process parameters, and equipment design. To tackle this issue, Barrasso et al. (2015) [103] devised a multi-scale modelling methodology that integrates a multi-dimensional PBM with simulations using the DEM. The particle size distribution (PBM) monitors the changes in particle size, liquid content, and porosity distributions within each compartment of the total suspended gas (TSG). On the other hand, the discrete element method (DEM) offers essential data on residence time, collision frequencies, and particle velocities. The ability to connect in both directions enables the systematic assessment of rate expressions that govern the important processes of granulation, including nucleation, aggregation, breakage, and consolidation.

The qualitative assessment of the prediction capability of the linked PBM-DEM model was conducted through the simulation of eight distinct screw configurations, each characterised by various numbers and offset angles of kneading elements (KEs). El Hagrasy and Litster (2014) [10] conducted a comparative analysis between the simulated trends and the experimental observations. The model effectively represented the primary granulation mechanisms seen in various screw elements, wherein nucleation was found to be predominant in the conveying elements and breaking was observed to be prominent in KEs. Higher residence periods and enhanced granule breakup were seen with an increase in the KE offset angle, although a larger number of KEs led to smaller granule sizes and lower fines. The model also forecasted enhanced liquid dispersion through heightened fragmentation and stratification in KEs. The results of this study align with the observed experimental patterns, demonstrating the model’s capacity to interpret the impact of screw design on granulation behaviour and product characteristics in a qualitative manner.

Notwithstanding the encouraging outcomes, certain disparities were observed between the simulated and experimental findings, indicating the necessity for enhancements in the model. For example, the model failed to consider the occurrence of fines during breaking events and exhibited a limited comprehension of the intricate mechanisms behind the distribution of fragment sizes. The significance of future research lies in its ability to enhance the mechanistic foundation of the model and integrate more accurate material attributes, hence improving its quantitative predictive capacity.

Barrasso et al. [104] introduced a multi-scale modelling framework that highlights the potential of integrating PBM and DEM simulations to enhance QbD in the field of pharmaceutical production. This technique has the capacity to be a helpful tool for model-driven design and optimisation of TSG processes by offering valuable insights into the intricate interactions among screw arrangement, granulation mechanisms, and product quality. With the advancement of the mechanistic comprehension of TSG and the emergence of more sophisticated computational methodologies, it is anticipated that the integrated PBM-DEM model will assume a progressively significant function in the advancement of resilient and effective continuous manufacturing procedures for solid oral dosage forms.

### 4.4. Future Directions and Emerging Trends

The twin-screw granulation simulation field is constantly advancing, offering several prospects for additional research and innovation. An emerging approach is the fusion of machine learning techniques with mechanistic models. Shirazian et al. (2017) [109] investigated the application of artificial neural networks (ANNs) in conjunction with PBM to forecast the distribution of granule sizes in a twin-screw granulation process [99]. The hybrid ANN and PBM methodology shows enhanced precision in comparison to the conventional PBM method alone. Similarly, Ismail et al. (2019) [8] devised a deep learning model to forecast granule characteristics and enhance process parameters in twin-screw granulation [7].

There is a growing interest in the creation of hybrid models that integrate several simulation methodologies. In their study, Hasanpoor et al. (2021) [110] utilised a com-bination of computational fluid dynamics (CFD), DEM, and PBM to examine the interaction between fluid flow and particle movement in a fluidised bed granulation [4]. The CFD-DEM model provided valuable insights into the mixing behaviour and mechanisms involved in the production of granules. This is an approach that could be implemented in future twin-screw granulation simulations.

Twin-screw granulation has become essential because of the growing implementation of continuous manufacturing, necessitating the need for real-time monitoring and control. In their study, Verstraeten and Maxim (2018) [111] combined process analytical technology (PAT) with a control framework based on population balance modelling (PBM) to enable the real-time monitoring and optimisation of a continuous twin-screw granulation process [5]. The utilisation of PAT techniques, such as near-infrared spectroscopy and Raman spectroscopy, allowed for the acquisition of real-time information regarding the characteristics of granules, thereby facilitating the implementation of adaptive process control.

Twin-screw granulation is an intricate procedure that can be enhanced by utilising flow sheet simulation with mechanistic models to forecast the distribution of granule size and moisture content. Arthur, Sekyi, Rahmanian, and Pu [34] utilised process modelling software, specifically the gPROMS gFP flowsheet simulation software (https://onlinelibrary.wiley.com/doi/full/10.1002/ceat.202200539, accessed on 18 May 2024), which employs a population balance model, to evaluate the impact of screw arrangement and liquid-to-solid ratio on the distribution of granule sizes. This method demonstrates potential in forecasting granule characteristics; nevertheless, it is necessary to consistently scrutinise and enhance the rate kernels to guarantee precise and replicable outcomes.

The population balance model employed in flow sheet modelling considers the different rate processes associated with twin-screw granulation, including nucleation, growth, breakage, and consolidation. By integrating these mechanistic models, the simulation may offer valuable insights into the granulation process and accurately forecast the ensuing distribution of granule sizes. The granulation process is influenced by two essential characteristics: the liquid-to-solid ratio and the screw arrangement. The impacts of these parameters can be analysed using simulation.

Although flow sheet modelling is a valuable tool for comprehending and enhancing twin-screw granulation, it is crucial to recognise its limitations and areas that can be improved. The precision of the simulation relies on the calibre of the rate kernels employed to depict the granulation process. The rate kernels are mathematical formulas that describe the rates at which different granulation methods occur. The regular analysis and improvement of these rate kernels are essential to guarantee that they precisely depict the many interactions inside the granulation process.

Future study should prioritise enhancing the comprehension of the underlying mechanics in twin-screw granulation and creating more precise and resilient rate kernels. This can be accomplished by employing a blend of empirical investigations, sophisticated characterisation methodologies, and mathematical simulation. By improving the rate kernels and verifying the simulation results with experimental data, the predictive power of flow sheet simulation can be strengthened, resulting in more effective process development and optimisation.

Ultimately, the advancement of twin-screw granulation simulation depends on the use of sophisticated modelling techniques, multi-scale methodologies, hybrid models, and real-time monitoring and control. The research landscape will be influenced by sustainability considerations and standardisation attempts. Effective collaboration among researchers, software developers, and industry stakeholders is crucial for fully harnessing the potential of simulations in optimising twin-screw granulation operations.

## 5. Conclusions

Overall, this literature review has conducted a thorough examination of the present condition of modelling and simulation techniques employed for the investigation of twin-screw granulation (TSG), a vital ongoing manufacturing procedure in the pharmaceutical sector. The review discussed the basic principles of wet granulation, the unique benefits and mechanisms of TSG, and the many methods employed to simulate and analyse the intricate interactions among equipment design, process parameters, and material properties in TSG. Discrete element method (DEM) simulations are now widely used to gain a detailed understanding of the particle-level processes involved in mixing, shearing, and wetting that control the formation and development of granules in TSG. Recent studies have shown that DEM models may accurately anticipate the impact of screw arrangement, barrel fill level, particle shape, and cohesiveness on granule characteristics and process performance. The use of GPUs in computers has made it possible to perform large-scale, computationally efficient discrete element method (DEM) simulations of TSG. The incorporation of physically based contact and bonding models, including the Timoshenko beam bond model and the JKR model, has enhanced the ability of DEM to accurately simulate intricate phenomena such as agglomeration breaking and cohesive mixing.

Population balance modelling (PBM) is a crucial method used to simulate the changes in granule size distributions and other characteristics in TSG. Novel multi-dimensional PBMs that incorporate granule size, liquid content, porosity, and composition have been created and verified using experimental data. Compartmental and regime-separated population balance models (PBMs) have proven to be successful in accurately representing the specific granulation mechanisms occurring in the various regions of the TSG barrel. Research in the field of physically based kernels for aggregation, breaking, and other rate processes is now underway, aiming to enhance the predictive capabilities of PBMs. The combined utilisation of DEM and PBM frameworks has yielded encouraging outcomes, facilitating the reciprocal transfer of information between particle-level and bulk-level models. These multi-scale techniques offer a comprehensive understanding of TSG and allow for the systematic assessment of rate expressions that influence the crucial granulation mechanisms.

Although there have been notable improvements in the modelling and simulation of TSG, there are still obstacles and potential areas for further investigation in future study. These advancements encompass the creation of more detailed kernels that connect model parameters to material properties and process conditions, the integration of particle wetting and liquid distribution phenomena, the utilisation of advanced characterisation techniques such as NIR and Raman spectroscopy to calibrate and validate models, and the implementation of high-performance computing for large-scale, predictive simulations.

As the pharmaceutical industry progressively adopts continuous manufacturing and quality by design (QbD) concepts, the incorporation of modelling and simulation tools into the TSG process development and optimisation workflow will become more crucial. Experimental studies, computational modelling, and commercial applications are highly effective tools that can be utilised. These methods provide useful insights and potential for optimisation in several sectors. There is significant potential to enhance the comprehension and efficiency of twin-screw granulation. This can result in improved product quality, reduced development time and cost, and increased process flexibility in the production of solid oral dosage forms.

## Figures and Tables

**Figure 1 pharmaceutics-16-00706-f001:**
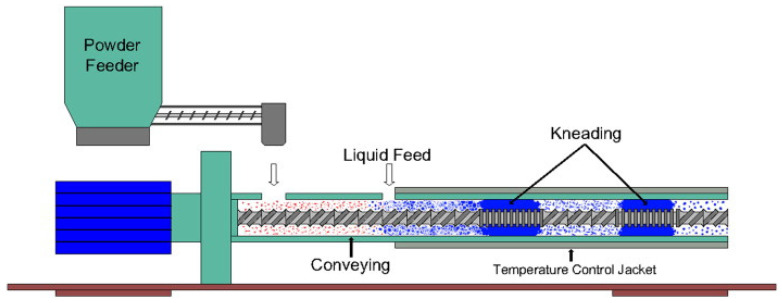
A schematic of a continuous twin-screw granulation [2].

**Figure 2 pharmaceutics-16-00706-f002:**
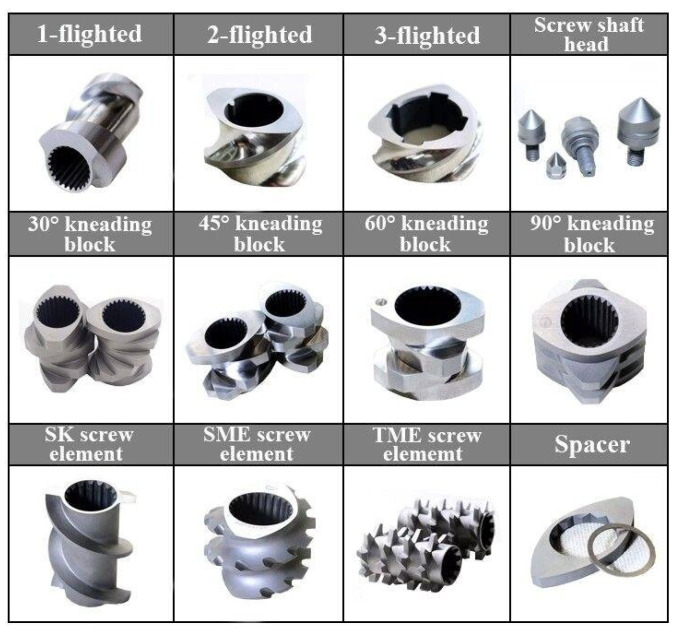
Twin-screw element, from Aftab (2018) [31].

**Figure 3 pharmaceutics-16-00706-f003:**
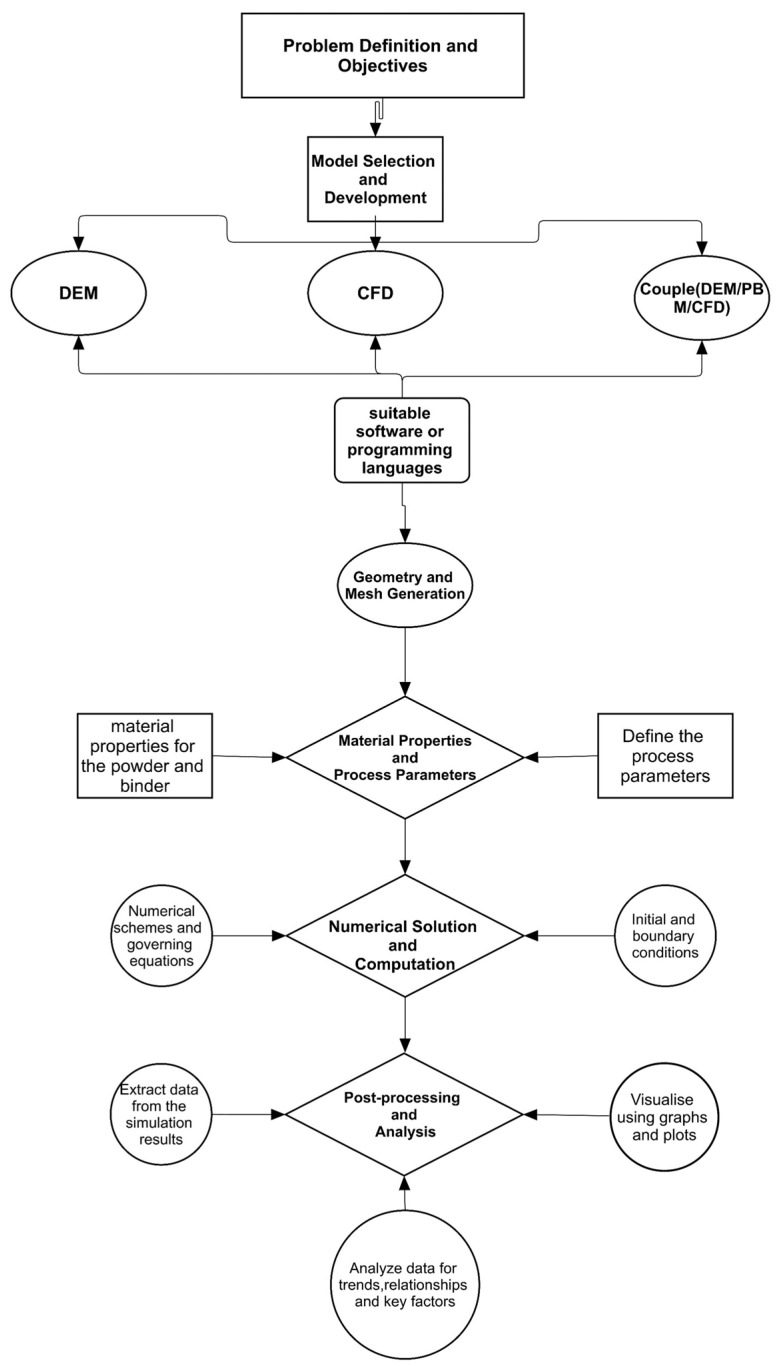
Schematic of simulation workflow.

**Table 1 pharmaceutics-16-00706-t001:** Simulation techniques and their strengths and limitations.

Simulation Techniques	Strength	Limitation
Discrete element method (DEM)	Offers comprehensive data on the development and enlargement of granules at the particle level [70]. Capable of simulating intricate particle geometries and interactions [71]. Enables understanding of mixing, shear, and compaction behaviour [72].	Computationally demanding, particularly for systems of significant size [73]. Depends on precise material characteristics and contact models [74]. Difficulties in simulating cohesive particles and wet granulation [75].
Population balance modelling (PBM)	Forecasts the progression of granule size distribution and other characteristics over a period of time [76]. Computationally efficient as compared to discrete approaches [77]. Can combine numerous modes of granulation, including aggregation, breaking, and consolidation [78].	Calibration is necessary using experimental data [79]. Constrained in managing intricacies and exchanges at the particle level [9]. Assumes uniform blending and may not account for specific variations in a particular area [79].
Computational fluid dynamics (CFD)	The software replicates the movement of fluids, the transfer of heat, and the transportation of substances within the granulator [80]. Methods (DEM) to simulate multiphase flows [81]. Accounts for the impact of equipment design and operational circumstances. Can be combined with particle-based methods (PBM) or discrete elements [82].	Proficiency in granulator geometry and mesh production is necessary [83]. Assumes a continuous representation and may not accurately depict the impacts of discrete particles [84]. High-resolution simulations can be computationally demanding [85].

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
