# Peer review of "Process Simulation of Twin-Screw Granulation: A Review"

_pharmaceutics, 2024, doi:10.3390/pharmaceutics16060706_

Round 1
Reviewer 1 Report
Comments and Suggestions for Authors
Dear Authors,
thank you for the positive review of TSG. I have some comments regarding the first part of your contribution, but the second part appears to be very informative and well-explained. I regret that I couldn't find any figures. Please double-check.
Reviewer

Author Response
- You are distinguishing between binder viscosity and viscosity of binder liquid. It is not clear, liquid viscosity depends on the binder concentration. Pls clarify!
Response:
Thanks for spotting this, this has been corrected to mean the viscosity of the liquid binder.
- inside TSG, add!
Response:
Thanks for spotting this, the whole paragraph has been rewritten, improved and restructured to address these concerns.
- This is not clear; viscosity is not able to produce a lot of bonds. If, then the polymers that are used in the binding liquid. With high viscosity might be present other phenomena: decreased spreadability of high viscosity granulating liquid. Did you consider this issue?
Response:
Thanks for spotting this, the whole paragraph has been rewritten, improved and restructured to address these concerns
- Has?
Response:
Do not know what this is implying
- Is than the filling rate not relevant?
Response:
The filling rate is an important parameter but has not been the focus of this section, however, the barrel fill levels has been discussed with respect to screw spead.it is understandable that the barrel fill level and the fill rate are closely related.
- which polymers are volatile? If something is volatile than the liquid, i.e. water or water mixtures. Pls clarify
Response:
Thanks for this, this has been corrected to “liquid binders composed of polymers which are of high volatility”.
- If I understand correctly the whole problem, then the temperature will never exceed 100 C, as water is granulating liquid. With organic liquids, when used, the temperature will be even lower. Pls comment.
Response:
Well, that maybe correct but higher RSD means material staying longer in the granulator thus, requiring more energy which tends to increase the temperature of the equipment due to mechanical action of the screws.
- Not found in the manuscript!
Response:
Thanks for spotting this, this has been corrected for in the updated version.
- not clear. What should be understood under fluid in this case.
Response:
Thanks for spotting this, this has been corrected to “two granulating materials”.
Reviewer 2 Report
Comments and Suggestions for Authors
This paper reviews applications of simulation to twin-screw granulators generally used in pharmaceutical industries. It discusses the topic from the point of view of process operations, modelling principles and simulation experiences. This is an extensive review (76 references), a well-written paper that deserves publication after some essential corrections.
Comments:
Title. (1) “Twin-screw granulator” of “twin screw granulator”? Both are used in this paper. as also in the literature. Use one form, only.
Keywords: (2) Only one simulation technology is mentioned; Discrete Element Method (DEM). The paper concerns also with other methods.
Introduction: (3) Introduction starts with an abbreviation (TSG). It should be opened here. (4) Is the long list of process variables taken from reference [2]? If so, it is better to put the reference earlier “In addition are studies to comprehend the equipment and screw geometries such as [2]”. (5) Introduction is also quite short and needs some necessary additions: What is the research target of this paper? What are its contributions? How was the review carried out? How were the references found out and chosen from the big group of potential ones? What do the authors think about the coverage of their review?
(6) References need some more care. Where are [1] and [17] published? Title of [6].
Author Response
- (1) “Twin-screw granulator” of “twin screw granulator”? Both are used in this paper. as also in the literature. Use one form, only.
Response:
Thanks for spotting this, these have been corrected to Twin-screw granulator.
- Keywords: (2) Only one simulation technology is mentioned; Discrete Element Method (DEM). The paper concerns also with other methods.
Response:
Thanks for spotting this, the keywords have been updated in the latest version to include other simulation techniques
- Introduction: (3) Introduction starts with an abbreviation(TSG). It should be opened here.
Response:
Thanks for spotting this, this has been corrected for in the updated version
- (4) Is the long list of process variables taken from reference [2]? If so, it is better to put the reference earlier “In addition are studies to comprehend the equipment and screw geometries such as [2]
Response:
Thanks for spotting this, this has been corrected for to be placed at where you have suggested in the updated version
- (5) Introduction is also quite short and needs some necessary additions: What is the research target of this paper? What are its contributions? How was the review carried out? How were the references found out and chosen from the big group of potential ones? What do the authors think about the coverage of their review?
Response:
Thanks for this, the latest manuscript has been updated to address these concerns
- (6) References need some more care. Where are [1] and [17] published? Title of [6]
Response:
Thanks for spotting this, this has been corrected in the updated version.
Reviewer 3 Report
Comments and Suggestions for Authors
The comprehensive review on twin-screw granulation processes is well-structured and informative. It effectively highlights the significance of this process in various industries, particularly in pharmaceuticals, for producing granules with controlled properties.
Here are some comments and suggestions for improvement:
The review is well-written and organized. However, to enhance clarity, the authors must consider providing a brief introductory paragraph outlining the importance and scope of twin-screw granulation before diving into the discussion of simulation techniques. Also, it must be pointed out the importance in powder processing industries and pharmaceuticals.
The review provides a good overview of simulation techniques employed in studying twin-screw granulation processes. However, it could benefit from a deeper discussion of specific case studies or applications where these techniques have been successfully utilized. By providing examples would enrich the reader's understanding and could demonstrate the practical relevance of the discussed methods.
While the review discusses various simulation techniques, it would be valuable to include a more critical analysis of their strengths and limitations. (as a Table)
The Equations in the manuscript are not well written according to Journal requests.
The review did not mention future research directions. The authors must consider discussing emerging trends or areas of innovation in twin-screw granulation simulation, such as the integration of machine learning techniques or the development of hybrid models combining multiple simulation methods. Add as a subchapter.
By incorporating figures or diagrams illustrating key concepts, such as the twin-screw granulator design or the simulation workflow, could enhance the clarity and accessibility of the review, particularly for readers less familiar with the topic.
The authors must ensure that the review includes references to the most recent literature (only 1 reference from 2023 is included and 2 from 2022) in the field of twin-screw granulation simulation to provide the latest developments and advancements in the area.
Overall, the present review provides a valuable resource for researchers interested in twin-screw granulation simulation. By addressing the above-mentioned points, the review can further enhance its impact and utility in the field.
Comments on the Quality of English LanguageSome minor English corrections are required.
Author Response
- The review is well-written and organized. However, to enhance clarity, the authors must consider providing a brief introductory paragraph outlining the importance and scope of twin-screw granulation before diving into the discussion of simulation techniques. Also, it must be pointed out the importance in powder processing industries and pharmaceuticals.
Response:
The updated version of the manuscript has factored in this comment and has provided a brief introductory paragraph.
- The review provides a good overview of simulation techniques employed in studying twin-screw granulation processes. However, it could benefit from a deeper discussion of specific case studies or applications where these techniques have been successfully utilized. By providing examples would enrich the reader's understanding and could demonstrate the practical relevance of the discussed methods.
Response:
Respectful, this comment is already addressed in the manuscript. The section 4 of the manuscript elucidates how these techniques and approaches have been implemented in literature and provides enough examples from literature.
- While the review discusses various simulation techniques, it would be valuable to include a more critical analysis of their strengths and limitations. (as a Table)
Response:
Thanks for this. This has been included in the updated version of the manuscript.
- The review did not mention future research directions. The authors must consider discussing emerging trends or areas of innovation in twin-screw granulation simulation, such as the integration of machine learning techniques or the development of hybrid models combining multiple simulation methods. Add as a subchapter.
Response:
Thanks for this, this has been factored in the latest update in section 4.4.
- By incorporating figures or diagrams illustrating key concepts, such as the twin-screw granulator design or the simulation workflow, could enhance the clarity and accessibility of the review, particularly for readers less familiar with the topic.
Response:
Thanks for this, diagrams of the twin-screw granulator, screw element and simulation workflow have been included in the updated version of the manuscript.
- The authors must ensure that the review includes references to the most recent literature (only 1 reference from 2023 is included and 2 from 2022) in the field of twin-screw granulation simulation to provide the latest developments and advancements in the area.
Response:
As difficult this may be at this stage of the paper, we have however included some recent references in the updated version of this manuscript.
Round 2
Reviewer 2 Report
Comments and Suggestions for Authors
I am delighted to see that the authors have been able to take into account my comments to the previous version. I recommend the publication of the paper.
Reviewer 3 Report
Comments and Suggestions for Authors
I agree with publication.
Best regards!